# Sharp Optimality of Simple, Plug-in Estimation
# of the Fisher Information of a Smoothed Density

**Subhodh Kotekal** [1]

## Abstract

Given independent and identically distributed data from a compactly supported, $\alpha$-Hölder density $f$, we study estimation of the Fisher information of the Gaussian-smoothed density $f * \varphi_t$, where $\varphi_t$ is the density of $N(0, t)$. We derive the minimax rate including the sharp dependence on $t$ and show some simple, plug-in type estimators are optimal for $t > 0$, even though extra debiasing steps are widely employed in the literature to achieve the sharp rate in the unsmoothed ($t = 0$) case. Due to our result's sharp characterization of the scaling in $t$, plug-in estimators of the mutual information and entropy are shown to achieve the parametric rate by way of the I-MMSE and de Bruijn's identities.

## 1. Introduction

Due to various recent motivations from machine learning (Tishby & Zaslavsky, 2015; Shwartz-Ziv & Tishby, 2017; Goldfeld et al., 2019; Song et al., 2021; Gabrié et al., 2018) and information theory (Goldfeld et al., 2020b; Goldfeld & Greenewald, 2020), the classical topic of the estimation of some fundamental statistical functionals of a probability density has seen renewed interest, now in the context of Gaussian smoothed distributions. In this article, we establish the minimax rate of estimating the Fisher information (also known as the Stein information) of a smoothed density $f * \varphi_t$ where $\varphi_t$ denotes the density of $N(0, t)$,

$$\mathcal{I}(f * \varphi_t) = \int_{-\infty}^{\infty} \frac{(f * \varphi_t)'(x)^2}{(f * \varphi_t)(x)} \, dx, \qquad (1)$$

given i.i.d. data from a compactly supported, $\alpha$-Hölder density $f$ with $\alpha \geq 1$. A plug-in type estimator is shown to achieve the optimal rate when $t > 0$, even though it

[1]Department of Statistics, The University of Chicago, Chicago IL, USA. Correspondence to: Subhodh Kotekal <skotekal@uchicago.edu>.

*Proceedings of the 42ⁿᵈ International Conference on Machine Learning*, Vancouver, Canada. PMLR 267, 2025. Copyright 2025 by the author(s).

is widely known plug-in estimators are suboptimal in the unsmoothed case ($t = 0$) (Laurent, 1997; Birgé & Massart, 1995). Furthermore, the minimax rate turns out to be slower for small $t > 0$; low amounts of Gaussian smoothing makes estimation of the Fisher information a harder problem compared to no smoothing, which is curious as Gaussian smoothing was proposed (Goldfeld et al., 2020b) in part to alleviate other statistical difficulties. Once $t > 0$, however, the minimax rate is shown to be decreasing in $t$, and there is a critical level of smoothing above which the problem is easier than if unsmoothed. The smoothed and unsmoothed settings are qualitatively different as the target density's support is, respectively, either unbounded or bounded. The estimation theory is materially affected since the Fisher information of $f * \varphi_t$ need not converge to that of $f$ as $t \to 0$; in fact, it may even diverge (see Remark 1.2). As a consequence of our results, the I-MMSE and de Bruijn identities (Guo et al., 2005; Stam, 1959) can be used to show that simple plug-in type estimators achieve the parametric rate for estimating the mutual information and the entropy, which are functionals of particular interest from the perspective of the Information Bottleneck theory (Shwartz-Ziv & Tishby, 2017; Tishby & Zaslavsky, 2015; Goldfeld et al., 2019; 2020b). Furthermore, our error bounds for the mutual information and the entropy do not blow up as $t \to 0$, which affected previous bounds in the literature (Goldfeld et al., 2020b).

Estimation of the Fisher information and other integral functionals are natural problems with widely appreciated motivations such as uncertainty quantification, estimation of information-theoretic divergences, and model selection. Hence, the unsmoothed case is, unsurprisingly, a well-studied problem with a long history which precludes a comprehensive review (Laurent, 1997; Donoho, 1988; Bhattacharya, 1967; Cao et al., 2020; Laurent, 1996; Birgé & Massart, 1995; Bickel & Ritov, 1988; Kandasamy et al., 2015) (see (Laurent, 1997; Donoho, 1988; Cao et al., 2020; Bhattacharya, 1967) for specific results concerning the Fisher information). The program of considering the smoothed setting was recently proposed in the work of Goldfeld et al. (2020b). It is classically known that, given $n$ i.i.d. data points from a $d$-dimensional distribution, though the empirical measure converges to the data-generating distribu-

tion in various senses, the convergence rate can be exponentially slow in the dimension (e.g. $n^{-1/d}$ in Wasserstein-1 distance for $d \geq 3$ (Dudley, 1968)). Motivated to circumvent this curse of dimensionality, Goldfeld et al. (2020b) suggest considering convergence of the empirical measure to the data-generating distribution after smoothing both with a Gaussian; they show the rate (in Wasserstein-1 distance, $\chi^2$-divergence, and total variation distance among others) drastically improves to the dimension-independent, parametric rate $n^{-1/2}$ (with prefactors depending on the dimension and the Gaussian's variance). Adopting the smoothing proposal, a growing literature has emerged including asymptotic distributional results for various smoothed distances (Goldfeld et al., 2024; Sadhu et al., 2022) and the development of the Gaussian-smoothed approach to optimal transport (Zhang et al., 2021; Goldfeld & Greenewald, 2020; Mena & Niles-Weed, 2019; Goldfeld et al., 2020a; Chewi et al., 2024; Ding & Niles-Weed, 2022; Chen & Niles-Weed, 2022).

Aside from convergence reasons, Goldfeld et al. (2020b) are also motivated by the Information Bottleneck theory (Shwartz-Ziv & Tishby, 2017; Tishby & Zaslavsky, 2015) in deep neural networks; a central quantity is the mutual information between the input feature vector and the hidden activation vector. Adopting a framework (Goldfeld et al., 2019) which relates this mutual information to an additive white Gaussian noise channel, estimation of this mutual information amounts to estimating the entropy of a smoothed density $f * \varphi_t$. Goldfeld et al. (2020b) establish the convergence rate for the plug-in estimator (entropy of the smoothed empirical measure). Given the connection of all these functionals, estimation of the Fisher information under Gaussian smoothing is of clear interest. The smoothed Fisher information has also been recently shown to be an essential quantity in the finite-sample analysis of mean estimation; we point the reader to Appendix H for a discussion of this recent line of results.

### 1.1. Related work

In the unsmoothed case ($t = 0$), a detailed estimation theory for the Fisher information is available. Laurent (1997) writes the Fisher information as $\int_{-1}^{1} (f')^2/f = \int_{-1}^{1} \Upsilon(f, f')$ with $\Upsilon(u, v) = v^2/u$, then proposes to use preliminary estimators $\hat{f}$ and $\hat{f}'$ to obtain the plug-in estimator $\int_{-1}^{1} \Upsilon(\hat{f}, \hat{f}')$, which is then debiased through an estimator of the error $\int_{-1}^{1} \Upsilon(f, f') - \Upsilon(\hat{f}, \hat{f}')$. The error estimator estimates the first two terms in the Taylor expansion of $\Upsilon$ at the point $(\hat{f}, \hat{f}')$ in the direction of $(f, f')$. Assuming $f$ belongs to an Sobolev space of index $\alpha > 1$, is compactly supported on, say, $[-\pi, \pi]$, satisfies some periodicity conditions, and is bounded below by a positive constant, Laurent (1997) shows the proposed estimator achieves the (squared error) rate $\frac{1}{n} + n^{-8(\alpha-1)/(4\alpha+1)}$, which matches the minimax lower

bound proved earlier by Birgé & Massart (1995). Moreover, when $\alpha > 9/4$ and the parametric rate $1/n$ is achieved, Laurent also shows the proposed estimator is semiparametrically efficient. The methodology and results developed in (Laurent, 1997) apply generally to integral functionals of the form $\int \Upsilon(f, f', \ldots, f^{(k)})$, though certain growth/decay conditions on $f$ and its derivatives are imposed to ensure $\Upsilon$ is a smooth function on the relevant domain (e.g. $f$ assumed bounded below by a constant so that the gradient of $\Upsilon(u, v) = v^2/u$ is bounded when $u = f(x)$).

These conditions preclude the theory of Laurent (1997) (and also (Laurent, 1996; Birgé & Massart, 1995)) from covering the smoothed case since $f * \varphi_t$ has unbounded support and is thus not bounded below by a constant. In fact, the minimax estimation rate of certain functionals of $f$ itself can actually change if $f$ is allowed to be arbitrarily close to zero in some regions of its support. For example, the entropy is no longer a smooth functional and the minimax rate of estimating the entropy of $f$ in squared error deteriorates from $\frac{1}{n} + n^{-4\alpha/(4\alpha+1)}$ to $\frac{1}{n} + (n \log n)^{-\alpha/(\alpha+1)}$ if $f$ lies in a Lipschitz ball with smoothness index $\alpha \in (0, 2]$ (see (Han et al., 2020) for details). However, such results are worst-case statements, and estimation of functionals of $f * \varphi_t$ can be faster when $f$ fulfills some regularity conditions since there is special, specific structure in the form of Gaussian convolution. Indeed, Goldfeld et al. (2020b) show a plug-in estimator of the entropy achieves the parametric rate $\frac{1}{n}$ (with prefactors depending on $t$).

For estimation of the Fisher information of $f$, Bhattacharya (1967) proposes the estimator $\widehat{\mathcal{I}(f)} = \int_{-k_n}^{k_n} \hat{f}'(x)^2/\hat{f}(x) \, dx$ where $k_n \to \infty$ is a sequence of truncation hyperparameters and $\hat{f}', \hat{f}$ are kernel density estimators of $f', f$ respectively. Under various restrictive assumptions on the size of $f$ and on the score $f'/f$, an asymptotic analysis was given in (Bhattacharya, 1967) (see also (Dmitriev & Tarasenko, 1974)), and later nonasymptotic results were derived by Cao et al. (2020). The upper bound on the error of Bhattacharya's estimator obtained in (Cao et al., 2020) is quite slow (logarithmic in $n$), and so Cao et al. propose a modification which truncates the integrand in addition to the domain of integration. The proposed estimator is shown to achieve an error bound which is polynomial in $n$. Cao et al. (2020) then apply their estimator specifically to Gaussian smoothed densities of the form $f * \varphi_t$, and derive some convergence rates. However, the rates are quite rough in the sense they are only shown to decay polynomially in $n$ with the precise exponents unclear. Furthermore, no minimax lower bounds are offered and so the question of optimality remains unresolved.

## 1.2. Main contribution

The notation used throughout the paper are defined in Appendix G. The parameter space for the minimax theory is defined as follows. For $\alpha, L > 0$ define the $\alpha$-Hölder class of probability density functions,

$$
\begin{aligned}
\mathcal{H}_\alpha(L) = \Big\{ & f : [-1, 1] \to [0, \infty) : \int_{-1}^{1} f(\mu) \, d\mu = 1, \\
& f \text{ is } \lfloor \alpha \rfloor\text{-times differentiable on } (-1, 1), \\
& \sup_{|\mu| < 1} |f^{(k)}(\mu)| \leq L \text{ for } k = 0, 1, 2..., \lfloor \alpha \rfloor - 1, \\
& \text{and } \left| f^{(\lfloor \alpha \rfloor)}(\mu) - f^{(\lfloor \alpha \rfloor)}(\mu') \right| \leq L |\mu - \mu'|^{\alpha - \lfloor \alpha \rfloor} \\
& \text{for all } \mu, \mu' \in (-1, 1) \Big\}.
\end{aligned}
\tag{2}
$$

This article focuses on the setting where $L > 0$ is some large universal constant, and so the notational dependence on $L$ will be suppressed. We will also treat $\alpha > 0$ as a fixed universal constant; all constants introduced implicitly or explicitly can potentially depend on $\alpha$. Define the parameter space

$$
\begin{aligned}
\mathcal{F}_\alpha := & \\
& \left\{ f : \mathbb{R} \to [0, \infty) : \text{supp}(f) \subset [-1, 1], f|_{[-1,1]} \in \mathcal{H}_\alpha, \right. \\
& \left. \text{and } c_d \leq f(\mu) \leq C_d \text{ for } |\mu| \leq 1 \right\},
\end{aligned}
\tag{3}
$$

where $C_d, c_d > 0$ are some universal constants. Throughout the paper, we will assume $\alpha \geq 1$. Consequently, for $f \in \mathcal{F}_\alpha$, the derivative $f'$ exists and satisfies $\sup_{|\mu| < 1} |f'(\mu)| \lesssim 1$. This is already assumed in the unsmoothed case $t = 0$ since otherwise consistent estimation (in a minimax sense) is not possible (Laurent, 1997).

In (3), we assume the data-generating density $f$ is bounded below by a constant in (3). Since the minimax rate for estimating $\mathcal{I}(f)$ might depend on whether $f$ is bounded below or not (e.g. in analogy to entropy estimation discussed earlier (Han et al., 2020)), we imagine the minimax rate for estimating $\mathcal{I}(f * \varphi_t)$ may also depend on whether $f$ is bounded below by a constant or not, even though $f * \varphi_t$ always has unbounded support and gets arbitrarily close to zero. Since (3) is a canonical class in nonparametric statistics (Tsybakov, 2009) and is interesting in its own right, we proceed with this assumption for this first investigation; the case where $f$ is not bounded away from zero is left open for future work.

Let $\varphi_t(x) = \frac{1}{\sqrt{2\pi t}} e^{-\frac{x^2}{2t}}$ denote the probability density function of $N(0, t)$. For a density $f \in \mathcal{F}_\alpha$, denote $p(x, t) = (f * \varphi_t)(x)$ and its (spatial) derivative $\partial_x p(x, t) = (f * \varphi_t)'(x)$. Given i.i.d. data $\mu_1, ..., \mu_n \sim f$, the goal is to estimate the Fisher information $\mathcal{I}_t := \mathcal{I}(f * \varphi_t)$ given by (1). Our main contribution is to show that for $\alpha \geq 1$ and $t > 0$ such that $t \leq c$ or $t \geq C$ for some sufficiently small universal constant $c > 0$ and sufficiently large universal constant $C > 0$, the sharp minimax rate is

$$
\inf_{\widehat{\mathcal{I}}_t} \sup_{f \in \mathcal{H}_\alpha} E \left( \left| \widehat{\mathcal{I}}_t - \mathcal{I}_t \right| \right) \asymp \frac{1}{\sqrt{n} t^2} \wedge \frac{1}{\sqrt{n} t^{3/4}} \wedge \frac{n^{-\frac{\alpha}{2\alpha+1}}}{\sqrt{t}}.
\tag{4}
$$

*Remark* 1.1. The rate (4) essentially exhibits three regimes. In the very high noise regime $t \gtrsim 1$, the rate is of order $1/(\sqrt{n} t^2)$. In the high noise regime $n^{-2/(2\alpha+1)} \lesssim t \lesssim 1$, the rate is of order $1/(\sqrt{n} t^{3/4})$, and the rate specializes to $n^{-\alpha/(2\alpha+1)}/\sqrt{t}$ in the low noise regime $t \lesssim n^{-2/(2\alpha+1)}$. In Section 2, three separate estimators are proposed to address each regime separately; all are essentially of plug-in type.

*Remark* 1.2. The minimax rate blows up as $t \to 0$, and this is because the Fisher information itself blows up. Consider the simple example $f(\mu) = \frac{1}{2} \mathbb{1}_{\{|\mu| \leq 1\}}$. Note $\mathcal{I}_t = \int_{-\infty}^{\infty} (\partial_x p(x, t))^2 / p(x, t) \, dx = \frac{1}{2} \int_{-\infty}^{\infty} (\varphi_t(x+1) - \varphi_t(x-1))^2 / P\{|N(x, t)| \leq 1\} \, dx \gtrsim \int_{|x-1| \leq \sqrt{t}} \varphi_t(x-1)^2 \, dx$ where the last inequality follows from $P\{|N(x, t)| \leq 1\} \asymp 1$ for $|x| \leq 1 + C\sqrt{t}$ as $t < 1$. We have also used that $\varphi_t(x+1) \leq c \varphi_t(x-1)$ for $|x-1| \leq \sqrt{t}$ where $c < 1$ is a small universal constant, since $t$ is small. Consider $\int_{|x-1| \leq \sqrt{t}} \varphi_t(x-1)^2 \, dx \asymp \frac{1}{\sqrt{t}} \int_{|x-1| \leq \sqrt{t}} \frac{1}{\sqrt{t}} e^{-(x-1)^2/t} \, dx \asymp \frac{1}{\sqrt{t}} \to \infty$ as $t \to 0$. The conceptual reason for the blowup is that $f$ is supported on $[-1, 1]$ with a sharp discontinuity at the endpoints since $f(-1), f(1) \geq c_d$ yet $f(x) = 0$ for $|x| > 1$. On the other hand, for any $t > 0$ no matter how small, $f * \varphi_t$ is positive everywhere. Consequently, the derivative $(f * \varphi_t)'$ explodes near the endpoints as $t \to 0$, causing the Fisher information to blow up.

*Remark* 1.3. As worked out in (Laurent, 1997), the minimax rate at $t = 0$ (ignoring conditions like periodicity and others stipulated in (Laurent, 1997)) in absolute error is $\frac{1}{\sqrt{n}} + n^{-4(\alpha-1)/(4\alpha+1)}$ provided $\alpha > 1$. A quick comparison to (4) shows there exists some critical threshold of $t$ under which the estimation problem becomes harder and over which it becomes easier. This result provides a caveat to Gaussian smoothing, proposed in (Goldfeld et al., 2020b) to alleviate the curse of dimensionality, even in the one-dimensional setting.

*Remark* 1.4. Under Gaussian smoothing, the Fisher information $\mathcal{I}_t$ can be related to the mutual information $I(f; f * \varphi_t)$ as well as the entropies $h(f * \varphi_t)$ and $h(f)$ through the celebrated I-MMSE and de Bruijn identities (Guo et al., 2005; Stam, 1959). In Section 4, we show the parametric rate of convergence for estimating these functionals can be achieved through a simple plug-in of the rate-optimal Fisher information estimator, despite the blowup in the optimal rate of Fisher information estimation as $t \to 0$.

*Remark* 1.5. There is a gap in our results for $c < t < C$. Our minimax upper bound actually goes through for all

$t > 0$. Rather, it is our analysis of the minimax lower bound which introduces the slack. We conjecture the gap is an artifact of our proof, and that a sharper argument could show a $\frac{1}{\sqrt{n}}$ lower bound for $c < t < C$.

## 2. Minimax upper bound

As noted earlier, it turns out a simple plug-in strategy for estimating $\mathcal{I}_t$ is rate-optimal. Recall the notation $p(x, t) = (f * \varphi_t)(x)$ and $\partial_x p(x, t) = (f * \varphi_t)'(x)$ for the diffused density and its (spatial) derivative. Define the function $\Upsilon : (0, \infty) \times \mathbb{R} \to \mathbb{R}$ with

$$\Upsilon(u, v) = \frac{v^2}{u}. \tag{5}$$

Note $\mathcal{I}_t = \int_{-\infty}^{\infty} \Upsilon(p(x, t), \partial_x p(x, t)) \, dx$. By a plug-in type estimator, we mean an estimator of the form

$$\widehat{\mathcal{I}}_t = \int_{-\infty}^{\infty} \Upsilon\left(\widehat{p}(x, t), \widehat{\partial_x p}(x, t)\right) \, dx,$$

where $\widehat{p}$ and $\widehat{\partial_x p}$ are some estimators of $p$ and $\partial_x p$ respectively.

Section 2.1 and Section 2.2 describe plug-in type estimators for the high noise ($n^{-2/(2\alpha+1)} < t < 1$) and the low noise ($t \leq n^{-2/(2\alpha+1)}$) regimes respectively. Section 2.3 addresses the very high noise regime ($t \geq 1$), where a plug-in strategy is essentially employed but with a slight modification to achieve the sharp rate; it is thus presented last for expositional clarity.

### 2.1. High noise regime

In the high noise regime $n^{-\frac{2}{2\alpha+1}} < t < 1$, it is intuitive to estimate $p(x, t) = (f * \varphi_t)(x)$ with the unbiased estimator $\widehat{p}(x, t) = \frac{1}{n}\sum_{i=1}^{n} \varphi_t(x - \mu_i)$. It directly follows $\partial_x \widehat{p}(x, t) = \frac{1}{n}\sum_{i=1}^{n} \varphi_t'(x - \mu_i)$, which is unbiased for $\partial_x p(x, t)$. Though it is appealing to use unbiased estimators, it turns out some truncation is convenient. Define

$$\underline{\varepsilon}(x, t) := c_d \int_{-1}^{1} \varphi_t(x - \mu) \, d\mu, \tag{6}$$

$$\overline{\varepsilon}(x, t) := C_d \int_{-1}^{1} \varphi_t(x - \mu) \, d\mu, \tag{7}$$

and note $\underline{\varepsilon}(x, t) \leq p(x, t) \leq \overline{\varepsilon}(x, t)$ since $f \in \mathcal{F}_\alpha$ implies $c_d \leq f(\mu) \leq C_d$ for $|\mu| \leq 1$. Likewise, define

$$\underline{\varepsilon}'(x, t) :=$$
$$c_d \varphi_t(x + 1) - C_d \varphi_t(x - 1) - L \int_{-1}^{1} \varphi_t(x - \mu) \, d\mu, \tag{8}$$

$$\overline{\varepsilon}'(x, t) :=$$
$$C_d \varphi_t(x + 1) - c_d \varphi_t(x - 1) + L \int_{-1}^{1} \varphi_t(x - \mu) \, d\mu. \tag{9}$$

Here, recall $|f'(\mu)| \leq L$ for all $|\mu| < 1$ (recall $\alpha \geq 1$ is assumed throughout) as stated in (3). To motivate the definitions (8) and (9), consider integration by parts yields $\partial_x p(x, t) = (f * \varphi_t')(x) = \int_{-1}^{1} f(\mu)\varphi_t'(x - \mu) \, d\mu = f(-1)\varphi_t(x + 1) - f(1)\varphi_t(x - 1) + \int_{-1}^{1} f'(\mu)\varphi_t(x - \mu) \, d\mu$, and so $\underline{\varepsilon}'(x, t) \leq \partial_x p(x, t) \leq \overline{\varepsilon}'(x, t)$. Finally, define the truncated estimators

$$\widehat{p}^{\varepsilon}(x, t) := (\widehat{p}(x, t) \vee \underline{\varepsilon}(x, t)) \wedge \overline{\varepsilon}(x, t), \tag{10}$$

$$\widehat{\partial_x p}^{\varepsilon}(x, t) := (\partial_x \widehat{p}(x, t) \vee \underline{\varepsilon}'(x, t)) \wedge \overline{\varepsilon}'(x, t). \tag{11}$$

**Theorem 2.1.** *If $\alpha \geq 1$ and $t < 1$, then there exists $C = C(\alpha, L)$ such that*

$$\sup_{f \in \mathcal{F}_\alpha} E\left(\left|\widehat{\mathcal{I}}_t - \mathcal{I}_t\right|\right) \leq \frac{C}{\sqrt{n} t^{3/4}},$$

*where*

$$\widehat{\mathcal{I}}_t := \int_{-\infty}^{\infty} \Upsilon\left(\widehat{p}^{\varepsilon}(x, t), \widehat{\partial_x p}^{\varepsilon}(x, t)\right) \, dx,$$

*with $\widehat{p}^{\varepsilon}$ and $\widehat{\partial_x p}^{\varepsilon}$ given by (10) and (11) respectively.*

The optimality of a plug-in estimator is notable since such a strategy is *not* optimal (with some other conditions such as periodicity of $f$ and $f'$) for estimating the Fisher information $\mathcal{I}(f)$ of the unsmoothed density (Laurent, 1997). Instead, Laurent (1997) employs a second order debiasing strategy to achieve the minimax rate, and even the optimal asymptotic variance in the regime where the parametric rate is achieved.

### 2.2. Low noise regime

In the low noise regime $t \leq n^{-\frac{2}{2\alpha+1}}$, the diffused density and its derivative will be estimated via a kernel smoothing. Suppose we have access to, say, a kernel-based density estimator $\hat{f}$ such that

$$E(|\hat{f}(\mu) - f(\mu)|^2) \lesssim n^{-\frac{2\alpha}{2\alpha+1}} \text{ for all } |\mu| \leq 1,$$

$$E(|\hat{f}'(\mu) - f'(\mu)|^2) \lesssim n^{-\frac{2(\alpha-1)}{2\alpha+1}} \text{ for all } |\mu| < 1,$$

which is standard (Tsybakov, 2009; Giné & Nickl, 2016). Define the preliminary estimators

$$\widehat{p}(x, t) = (\hat{f} * \varphi_t)(x), \tag{12}$$

$$\widehat{\partial_x p}(x, t) = \hat{f}(-1)\varphi_t(x + 1) - \hat{f}(1)\varphi_t(x - 1)$$
$$+ \int_{-1}^{1} \hat{f}'(\mu)\varphi_t(x - \mu) \, d\mu. \tag{13}$$

The definition of $\widehat{\partial_x p}$ is motivated by noting that integration by parts implies the estimand can be written as $\partial_x p(x, t) = (f * \varphi_t')(x) = \int_{-1}^{1} f(\mu) \varphi_t'(x - \mu) \, d\mu = f(-1)\varphi_t(x + 1) -$

$f(1)\varphi_t(x-1) + \int_{-1}^{1} f'(\mu)\varphi_t(x-\mu)\,d\mu$. To construct a plug-in estimator of $\mathcal{I}_t$, we will truncate the preliminary estimators as in the high noise regime,

$$\widehat{p}^{\varepsilon}(x,t) = (\widehat{p}(x,t) \vee \underline{\varepsilon}(x,t)) \wedge \overline{\varepsilon}(x,t), \qquad (14)$$

$$\widehat{\partial_x p}^{\varepsilon}(x,t) = \left(\widehat{\partial_x p}(x,t) \vee \underline{\varepsilon}'(x,t)\right) \wedge \overline{\varepsilon}'(x,t). \qquad (15)$$

where the truncation levels are given by (6), (7), (8), and (9).

**Theorem 2.2.** *If $\alpha \geq 1$ and $t < 1$, then there exists $C = C(\alpha, L)$ such that*

$$\sup_{f \in \mathcal{F}_{\alpha}} E\left(\left|\widehat{\mathcal{I}}_t - \mathcal{I}_t\right|\right) \leq C\left(n^{-\frac{\alpha-1}{2\alpha+1}} + \frac{n^{-\frac{\alpha}{2\alpha+1}}}{\sqrt{t}}\right)$$

*where*

$$\widehat{\mathcal{I}}_t := \int_{-\infty}^{\infty} \Upsilon\left(\widehat{p}^{\varepsilon}(x,t), \widehat{\partial_x p}^{\varepsilon}(x,t)\right)\,dx,$$

*with $\widehat{p}^{\varepsilon}$ and $\widehat{\partial_x p}^{\varepsilon}$ given by (14) and (15) respectively.*

The dominating term in the upper bound established by Theorem 2.2 is $n^{-\alpha/(2\alpha+1)}/\sqrt{t}$ in the regime $t \leq n^{-2/(2\alpha+1)}$. In the high noise regime $n^{-2/(2\alpha+1)} < t < 1$, the rate $\frac{1}{\sqrt{n}t^{3/4}}$ achieved by the estimator in Theorem 2.1 is faster; in other words, Theorems 2.1 and 2.2 together imply the minimax upper bound $\frac{1}{\sqrt{n}t^{3/4}} \wedge \frac{n^{-\frac{\alpha}{2\alpha+1}}}{\sqrt{t}}$ for $t < 1$.

## 2.3. Very high noise regime

In the very high noise regime $t \geq 1$, we will slightly modify the plug-in strategy used in Sections 2.1 and 2.2. Consider $\partial_x p(x,t) = (f * \varphi_t')(x) = \int_{-1}^{1} -\frac{x-\mu}{t}\varphi_t(x-\mu)f(\mu)\,d\mu = -\frac{x}{t}p(x,t) + \frac{1}{t}\int_{-1}^{1}\mu\varphi_t(x-\mu)f(\mu)\,d\mu$. Therefore,

$$
\begin{aligned}
\mathcal{I}_t &= \int_{-\infty}^{\infty} \frac{(\partial_x p(x,t))^2}{p(x,t)}\,dx \\
&= \frac{1}{t^2}\int_{-\infty}^{\infty} x^2 p(x,t)\,dx \\
&\quad - \frac{2}{t^2}\int_{-\infty}^{\infty}\int_{-1}^{1} x\mu\varphi_t(x-\mu)f(\mu)\,d\mu\,dx \\
&\quad + \int_{-\infty}^{\infty}\Upsilon\left(p(x,t), \frac{1}{t}\int_{-1}^{1}\mu\varphi_t(x-\mu)f(\mu)\,d\mu\right)\,dx \\
&= \left(\frac{1}{t} - \frac{1}{t^2}\int_{-1}^{1}\mu^2 f(\mu)\,d\mu\right) \\
&\quad + \int_{-\infty}^{\infty}\Upsilon\left(p(x,t), \frac{1}{t}\int_{-1}^{1}\mu\varphi_t(x-\mu)f(\mu)\,d\mu\right)\,dx \\
&=: L_1 + L_2.
\end{aligned}
$$

The two terms will be estimated separately. The first two terms can be simply estimated through unbiased estimators,

$$\widehat{L}_1 := \frac{1}{t} - \frac{1}{nt^2}\sum_{i=1}^{n}\mu_i^2. \qquad (16)$$

$L_2$ is estimated via plug-in, namely

$$\widehat{L}_2 := \int_{-\infty}^{\infty} \Upsilon\left(\widehat{p}^{\varepsilon}(x,t), \widehat{q}^{\varepsilon}(x,t)\right)\,dx \qquad (17)$$

where $\widehat{p}^{\varepsilon}$ is given by (10) and

$$\widehat{q}^{\varepsilon}(x,t) :=$$
$$\frac{1}{t}\left(\left(\left(\frac{1}{n}\sum_{i=1}^{n}\mu_i\varphi_t(x-\mu_i)\right) \wedge \overline{\varepsilon}(x,t)\right) \vee -\overline{\varepsilon}(x,t)\right)$$
$$\qquad (18)$$

with $\overline{\varepsilon}(x,t)$ given by (7). The regularization in $\widehat{q}^{\varepsilon}(x,t)$ is applied since the target satisfies $\left|\int_{-1}^{1}\mu\varphi_t(x-\mu)f(\mu)\,d\mu\right| \leq \overline{\varepsilon}(x,t)$.

**Theorem 2.3.** *If $c > 0$, then for $t \geq c$, we have*

$$\sup_{f \in \mathcal{F}_{\alpha}} E\left(\left|\widehat{\mathcal{I}}_t - \mathcal{I}_t\right|\right) \lesssim \frac{1 + c^{-1/2}}{\sqrt{n}t^2},$$

*where $\widehat{\mathcal{I}}_t := \widehat{L}_1 + \widehat{L}_2$ given by (16) and (17).*

Theorems 2.1, 2.2, and 2.3 collectively establish the upper bound in (4).

## 3. Minimax lower bound

Theorem 3.1 states a minimax lower bound for the high and low noise regimes, which establishes the optimality of the plug-in methodology described in Sections 2.1 and 2.2.

**Theorem 3.1.** *There exists a sufficiently small universal constant $c > 0$ such that the following holds. If $\alpha \geq 1$, then*

$$\inf_{\widehat{\mathcal{I}}_t} \sup_{f \in \mathcal{F}_{\alpha}} E\left(\left|\widehat{\mathcal{I}}_t - \mathcal{I}_t\right|\right) \gtrsim \left(\frac{1}{\sqrt{n}t^{3/4}} \wedge \frac{n^{-\frac{\alpha}{2\alpha+1}}}{\sqrt{t}}\right)$$

*for $t \leq c$.*

The proof of Theorem 3.1 employs a two-point construction. However, standard constructions (Tsybakov, 2009) appear not to deliver the sharp minimax lower bound, particularly for $t \gtrsim n^{-2/(2\alpha+1)}$. Instead, we employ an idea introduced by Dou et al. (2024), who established sharp minimax lower bounds for score matching in score-based diffusion models (i.e. estimation of the score function $\partial_x \log p(x,t)$ of the diffused density).

Let $K : \mathbb{R} \to \mathbb{R}$ denote a symmetric (i.e. even) probability density function (i.e. $K \geq 0$ and $\int_{-\infty}^{\infty} K(x)\,dx = 1$) which

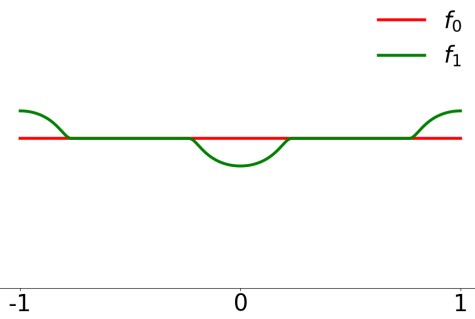

Figure 1. A cartoon of $f_0$ and $f_1$.

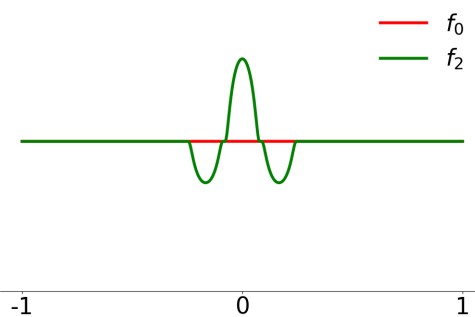

Figure 2. A cartoon of $f_0$ and $f_2$.

is supported on $[-1, 1]$ such that $K \in C^\infty(\mathbb{R})$, $||K||_\infty$ is bounded by a universal constant, $K$'s first $\lceil \alpha \rceil$ derivatives have $L_\infty$ norm bounded by a constant (which may depend on $\alpha$ and $L$), and $K(0) = \sup_{x \in \mathbb{R}} K(x)$. The two data-generating densities deployed in the proof are

$$f_0(\mu) = \frac{1}{2} \mathbb{1}_{\{|\mu| \le 1\}}, \tag{19}$$

$$f_1(\mu) = f_0(\mu) + g(\mu) \tag{20}$$

where

$$g(\mu) = \left( -\epsilon^\alpha K\left(\frac{\mu}{\rho}\right) + \epsilon^\alpha K\left(\frac{\mu-1}{\rho}\right) \right.$$
$$\left. + \epsilon^\alpha K\left(\frac{\mu+1}{\rho}\right) \right) \mathbb{1}_{\{|\mu| \le 1\}}. \tag{21}$$

Here, $0 < \epsilon \le \rho < 1$ are parameters to be tuned in the course of the proof. It is clear $f_0 \in \mathcal{F}_\alpha$. To check $f_1 \in \mathcal{F}_\alpha$, the Hölder condition must be verified. The constraint $\rho \ge \epsilon$ ensures $f_1$ is Hölder smooth. Traditional constructions (Tsybakov, 2009) do not introduce an extra free parameter $\rho$, but rather implicitly take $\rho = \epsilon$; the idea to introduce another tunable parameter is due to Dou et al. (2024) as noted earlier. Further, $f_1$ integrates to 1 since Lemma E.2 asserts that $g$ integrates to 0, and so provided $\epsilon \le \rho$ are chosen smaller than a sufficiently small universal constant, it is straightforward to verify $f_1 \in \mathcal{F}_\alpha$. In the course of the proof of Theorem 3.1, the optimal choices of the tuning parameters turn out to be $\rho \asymp \sqrt{t} \vee \epsilon$ and $\epsilon \asymp \left(\frac{1}{n\sqrt{t}}\right)^{\frac{1}{2\alpha}} \wedge n^{-\frac{2}{2\alpha+1}}$.

The intuition for the construction is similar to the intuition described in Remark 1.2 for the blowup of (4) as $t \to 0$. Namely, the endpoints of the interval $[-1, 1]$ are the problematic points which can cause large estimation error. Thus, $f_0$ and $f_1$ are chosen such that they are different near

the boundary of $[-1, 1]$. Since $f_1$ needs to integrate to 1 since it is a density, some mass needs to be displaced from somewhere else to compensate for the mass difference at the boundary; we thus take mass out near zero as shown in Figure 1.

Theorem 3.2 states a lower bound for the very high noise regime $t \gtrsim 1$ which matches the upper bound presented in Section 2.3.

**Theorem 3.2.** *There exists a sufficiently large universal constant $C > 0$ such that the following holds. If $\alpha \ge 1$, then*

$$\inf_{\widehat{\mathcal{I}}_t} \sup_{f \in \mathcal{F}_\alpha} E\left( \left| \widehat{\mathcal{I}}_t - \mathcal{I}_t \right| \right) \gtrsim \frac{1}{\sqrt{n} t^2}$$

*for $t \ge C$.*

A two-point construction is also used to prove Theorem 3.2. Let $\psi : \mathbb{R} \to \mathbb{R}$ denote a function such that $\psi \in C^\infty(\mathbb{R})$, $\psi$ is supported on $[-1, 1]$, $||\psi||_\infty$ is bounded by a universal constant, $\psi$'s first $\lceil \alpha \rceil$ derivatives have $L_\infty$ norm bounded by a constant (which may depend on $\alpha$ and $L$), $\int_{-\infty}^{\infty} \psi(\nu)\, d\nu = 0$, $\int_{-\infty}^{\infty} \psi(\nu)\nu\, d\nu = 0$, and $\left| \int_{-\infty}^{\infty} \psi(\nu)\nu^2\, d\nu \right| \ge c$ where $c > 0$ is some universal constant. The two-point construction we use are $f_0$ given by (19) and

$$f_2(\mu) = f_0(\mu) + \epsilon^\alpha \psi\left(\frac{\mu}{\rho}\right) \tag{22}$$

where $0 < \epsilon \le \rho < 1$ are parameters to be tuned. Figure 2 presents a cartoon of $f_0$ and $f_2$. It is straightforward to verify $f_2 \in \mathcal{F}_\alpha$ provided $\epsilon \le \rho$ are chosen smaller than a sufficiently small universal constant. It turns out the optimal choices are $\rho \asymp 1$ and $\epsilon \asymp n^{-\frac{1}{2\alpha}}$.

In the very high noise regime, $t$ happens to be large enough to smooth out the discontinuities of $f$ at the endpoints of $[-1, 1]$ that it does not cause fundamental slowdowns in the estimation rate. From the perspective of the lower bound

construction, it turns out we can select our two points $f_0$ and $f_2$ which actually agree near the boundary of $[-1, 1]$ and still obtain a sharp lower bound. Hence, we make the choice (22), where the perturbation $\mu \mapsto \epsilon^\alpha \psi(\mu/\rho)$ is localized near $0$ and does not affect the boundary. This kind of perturbation will be readily recognized as standard to the expert, excepting the aforementioned idea of Dou et al. (2024) to introduce $\rho$.

# 4. Estimation of mutual information and entropy: plug-in achieves the parametric rate

The additive Gaussian white noise channel is a canonical model in information theory and estimation theory. In this context, the Fisher information is related to other information-theoretic quantities of fundamental interest; we will focus on mutual information and entropy here. As a consequence of our results on Fisher information estimation, it can be shown through the I-MMSE (Guo et al., 2005) and de Bruijn's (Stam, 1959) identities (which are actually equivalent (Guo et al., 2005)) that a simple plug-in type estimator can estimate at parametric rate both the mutual information between a signal and its noisy measurement through an additive Gaussian white noise channel, as well as the entropy of the source distribution. The idea to plug in an estimator of the Fisher information into these identities has appeared before in the literature (Cao et al., 2020).

Suppose a clean signal $\mu \sim f$ is passed through an additive Gaussian noise channel to yield the measurement $X_t \mid \mu \sim N(\mu, t)$. Given i.i.d. data $\mu_1, ..., \mu_n \sim f$, consider the problem of estimating the mutual information $I(X_t; \mu)$. The I-MMSE identity (Guo et al., 2005) coupled with Brown's identity (Brown, 1971) relates the Fisher information of the smoothed density $f * \varphi_t$ to the mutual information of interest, as asserted by Theorem 4.1, whose proof is deferred to Appendix F.

**Theorem 4.1.** *For $t > 0$, we have the identity*

$$\mathcal{I}_t = \frac{1}{t}\left(1 + 2t\frac{d}{dt}I(X_t; \mu)\right). \qquad (23)$$

Since $I(X_t; \mu) \to 0$ as $t \to \infty$, it follows by the fundamental theorem of calculus that

$$I(X_t; \mu) = \frac{1}{2}\int_t^\infty \frac{1}{s} - \mathcal{I}_s \, ds.$$

Plugging in our estimator of the Fisher information (depending on the regime of $s$) and integrating yields the parametric rate.

**Theorem 4.2.** *Suppose $\alpha \geq 1$. Let $\widehat{\mathcal{I}}_s$ be the estimator in Theorem 2.3 for $s \geq 1$, Theorem 2.1 for $n^{-\frac{2}{2\alpha+1}} < s < 1$,*

*and Theorem 2.2 for $0 < s \leq n^{-\frac{2}{2\alpha+1}}$. Fix any $t > 0$. Define the estimator*

$$\widehat{I(X_t; \mu)} := \frac{1}{2}\int_t^\infty \frac{1}{s} - \widehat{\mathcal{I}}_s \, ds.$$

*If $t > 0$, then there exists $C = C(\alpha, L)$ such that*

$$\sup_{f \in \mathcal{F}_\alpha} E\left(\left|\widehat{I(X_t; \mu)} - I(X_t; \mu)\right|\right) \leq \frac{C}{\sqrt{n}}.$$

*Proof.* By (23) along with Theorems 2.3, 2.1, and 2.2, it directly follows $E(|\widehat{I(X_t; \mu)} - I(X_t; \mu)|) \lesssim C\int_t^\infty \frac{1}{\sqrt{n}s^2} \wedge \frac{1}{\sqrt{n}s^{3/4}} \wedge \frac{n^{-\frac{\alpha}{2\alpha+1}}}{\sqrt{s}} \, ds \lesssim \frac{C}{\sqrt{n}}\int_1^\infty \frac{1}{s^2} \, ds + \frac{C}{\sqrt{n}}\int_{n^{-\frac{2}{2\alpha+1}}}^1 \frac{1}{s^{3/4}} \, ds + Cn^{-\frac{\alpha}{2\alpha+1}}\int_t^{n^{-\frac{2}{2\alpha+1}}} \frac{1}{\sqrt{s}} \, ds \lesssim \frac{C}{\sqrt{n}} + Cn^{-\frac{\alpha+1}{2\alpha+1}}$. Since $\frac{\alpha+1}{2\alpha+1} > \frac{1}{2}$, the dominating term is $\frac{1}{\sqrt{n}}$. $\square$

Estimation of the entropy of the smoothed density follows immediately since

$$h(f * \varphi_t) = I(X_t; \mu) - h(X_t \mid \mu) = I(X_t; \mu) - \frac{1}{2}\log(2\pi e t)$$

by the well-known property $I(U; V) = h(U) - h(U \mid V)$ for continuous random variables $U$ and $V$. It is immediate to see that the parametric rate can be achieved, as we state in Corollary 4.3 without proof.

**Corollary 4.3.** *Suppose $\alpha \geq 1$. Let $\widehat{I(X_t; \mu)}$ be the estimator from Theorem 4.2 and define*

$$\widehat{h(f * \varphi_t)} := \widehat{I(X_t; \mu)} - \frac{1}{2}\log(2\pi e t).$$

*If $t > 0$, then there exists $C = C(\alpha, L)$ such that*

$$\sup_{f \in \mathcal{F}_\alpha} E\left(\left|\widehat{h(f * \varphi_t)} - h(f * \varphi_t)\right|\right) \leq \frac{C}{\sqrt{n}}.$$

The upper bound stated in Corollary 4.3 notably does not blow up as $t \to 0$. This is in stark contrast to the upper bound obtained by Goldfeld et al. (2020b) for the estimator $\widehat{h(f * \varphi_t)} = h\left(\frac{1}{n}\sum_{i=1}^n \delta_{\mu_i} * \varphi_t\right)$; the constant in (33) of Goldfeld et al. (2020b) is explicitly characterized in (63), where it is plainly seen to diverge as $t \to 0$.

Via de Bruijn's identity (Stam, 1959),

$$\mathcal{I}_t = 2\frac{d}{dt}h(f * \varphi_t), \qquad (24)$$

an estimator can also be directly constructed for the entropy of the unsmoothed data density $f$ (i.e. the case $t = 0$). The

fundamental theorem of calculus[1] yields

$$h(f) = h(f * \varphi_t) - \frac{1}{2} \int_0^t \mathcal{I}_s \, ds.$$

Again, it is straightforward to see a plug-in type estimator achieves the parametric rate, as stated in Corollary 4.4 without proof.

**Corollary 4.4.** *Suppose $\alpha \geq 1$. Let $\mathcal{I}_s$ denote the estimator of the Fisher information defined in Theorem 4.2 and $\widehat{h(f * \varphi_t)}$ denote the estimator defined in Corollary 4.3. Fix any $t > 0$. Define*

$$\widehat{h(f)} = \widehat{h(f * \varphi_t)} - \frac{1}{2} \int_0^t \widehat{\mathcal{I}}_s \, ds.$$

*Then there exists $C = C(\alpha, L)$ such that*

$$\sup_{f \in \mathcal{F}_\alpha} E\left( \left| \widehat{h(f)} - h(f) \right| \right) \leq \frac{C}{\sqrt{n}}.$$

# 5. Proof ideas

This section briefly sketches the essence of the proof strategies, which is to examine Taylor expansions of $\Upsilon$.

## 5.1. Upper bound: bounding the estimation error

To establish Theorems 2.1, 2.2, and 2.3, the estimation errors will be bounded through a Taylor expansion of $\Upsilon$ defined in (5). The proof of Proposition 5.1 is deferred to Appendix A.

**Proposition 5.1.** *If $(u, v), (u_0, v_0) \in (0, \infty) \times \mathbb{R}$, then*

$$\Upsilon(u, v) = \Upsilon(u_0, v_0) + \Gamma_{u_0, v_0}(u, v)$$

*where the remainder $\Gamma_{u_0, v_0}(u, v)$ satisfies*

$$|\Gamma_{u_0, v_0}(u, v)|$$
$$\leq C\left( \frac{(|v| \vee |v_0|)^2}{(|u| \wedge |u_0|)^2} |u - u_0| + \frac{|v| \vee |v_0|}{|u| \wedge |u_0|} |v - v_0| \right)$$

*where $C > 0$ is some universal constant.*

In Appendix C, Proposition 5.1 is employed to obtain bounds on the estimation error, by recognizing

$$\left| \widehat{\mathcal{I}}_t - \mathcal{I}_t \right|$$
$$= \left| \int_{-\infty}^\infty \Upsilon(\widehat{p}^\varepsilon(x, t), \widehat{\partial_x p}^\varepsilon(x, t)) - \Upsilon(p(x, t), \partial_x p(x, t)) \, dx \right|$$
$$\leq \int_{-\infty}^\infty |\Gamma_{p(x,t), \partial_x p(x,t)}(\widehat{p}^\varepsilon(x, t), \widehat{\partial_x p}^\varepsilon(x, t))| \, dx.$$

[1]The careful reader may be concerned about the continuity condition $h(f * \varphi_t) \to h(f)$ as $t \to 0$ needed in order to invoke the (second) fundamental theorem of calculus. The desired continuity has been used implicitly in the literature and might be considered folklore by some, but an explicit discussion and a proof can be found in Theorem 6.2 and Remark 10 of (Wang & Madiman, 2014) (see also (Rioul, 2017) for a discussion of other works containing this result).

Bounding the estimation error amounts to carefully bounding the remainder term $\Gamma$.

## 5.2. Lower bound: bounding the functional separation

As mentioned in Section 3, the lower bounds are proved through two-point constructions and applications of Le Cam's method. A crucial ingredient is to obtain lower bounds on separation of the Fisher informations of the two constructed densities. For discussion, let us first focus on bounding $|\mathcal{I}(f_1 * \varphi_t) - \mathcal{I}(f_0 * \varphi_t)|$ from below in the proof of Theorem 3.1.

From the upper bound perspective, the dominating term in the estimation error appears to be the linear term in the Taylor expansion of $\Upsilon$. Therefore, it is natural in the lower bound to pursue an argument asserting the order of the functional separation is dominated by this first order term. To carry out such an argument, the linear term will need to be computed explicitly, and the remainder must be argued to be of smaller order. Proposition 5.2 gives the expansion, and its proof is deferred to Appendix B.

**Proposition 5.2.** *If $(u, v), (u_0, v_0) \in (0, \infty) \times \mathbb{R}$, then*

$$\Upsilon(u, v) = \Upsilon(u_0, v_0) - \frac{v_0^2}{u_0^2}(u - u_0) + \frac{2v_0}{u_0}(v - v_0) + \Gamma_{u_0, v_0}(u, v)$$

*where the remainder satisfies*

$$|\Gamma_{u_0, v_0}(u, v)|$$
$$\leq C\left( \frac{|v|^2 \vee |v_0|^2}{|u|^3 \wedge |u_0|^3} |u - u_0|^2 + \frac{1}{|u| \wedge |u_0|} |v - v_0|^2 \right.$$
$$\left. + \frac{|v| \vee |v_0|}{|u|^2 \wedge |u_0|^2} |u - u_0||v - v_0| \right).$$

To illustrate the use of Proposition 5.2, consider it yields

$$\mathcal{I}(f_1 * \varphi_t) - \mathcal{I}(f_0 * \varphi_t)$$
$$\approx - \int_{-\infty}^\infty \frac{(\partial_x p_0(x, t))^2}{p_0(x, t)^2} (p_1(x, t) - p_0(x, t)) \, dx$$
$$+ 2 \int_{-\infty}^\infty \frac{\partial_x p_0(x, t)}{p_0(x, t)} (\partial_x p_1(x, t) - \partial_x p_0(x, t)) \, dx.$$

In Appendix B, the right hand side is carefully examined to yield a lower bound on the separation of the Fisher information. Jumping to the punchline, it is essentially shown

$$\mathcal{I}(f_1 * \varphi_t) - \mathcal{I}(f_0 * \varphi_t)$$
$$\approx \int_0^\infty \frac{\varphi_t(x - 1)^2}{p_0(x, t)} \left( 2g(1) - \frac{g(x, t)}{2p_0(x, t)} \right) dx$$

where $p_0(x, t) = (f_0 * \varphi_t)(x)$ and $g(x, t) = (g * \varphi_t)(x)$. The posterior expectation can be quickly recognized $\frac{g(x,t)}{2p_0(x,t)} = E(g(\mu) \,|\, X_t = x)$ in the Bayes model

$\mu \sim f_0$ and $X_t | \mu \sim N(\mu, t)$. The above display further motivates the choice of $g$ made in (21) and depicted in Figure 1, namely we wish to choose $g$ such that $2g(1) - E(g(\mu) | X_t = x)$ is bounded away from zero. A sufficient choice is to select $g$ which attains its maximum at $\{1, -1\}$ and is bounded away from zero here; this is congruent with the intuition offered in Section 3.

A similar Taylor expansion strategy is used in the proof of Theorem 3.2 for bounding $|\mathcal{I}(f_2 * \varphi_t) - \mathcal{I}(f_0 * \varphi_t)|$. However, it turns out some simple manipulation enables the use of the simpler expansion of Proposition 5.1. This can be related to the intuition offered in Section 3, which notes the endpoints of $[-1, 1]$ no longer cause a slowdown in the estimation rate since there is much smoothing in the very high noise regime. We defer the details to Appendix B.2.

## Acknowledgements

This research was supported by NSF Grant ECCS-2216912 and the Institute for Data, Econometrics, Algorithms, and Learning (IDEAL).

## Impact Statement

This paper presents work whose goal is to advance the field of Machine Learning. There are many potential societal consequences of our work, none which we feel must be specifically highlighted here.

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

# Appendices to "Sharp optimality of simple, plug-in estimation of the Fisher information of a smoothed density"

## A. Proofs: upper bound

Theorems 2.1 and 2.2 involve plug-in estimators of the form

$$\widehat{\mathcal{I}}_t = \int_{-\infty}^{\infty} \Upsilon(\widehat{p}^{\,\varepsilon}(x,t), \widehat{\partial_x p}^{\,\varepsilon}(x,t)) \, dx,$$

Before using Proposition 5.1 to bound the error, we first present its proof.

*Proof of Proposition 5.1.* Observe $\nabla \Upsilon(u,v) = \left(-\frac{v^2}{u^2}, \frac{2v}{u}\right)$. For $(u,v), (u_0, v_0) \in (0, \infty) \times \mathbb{R}$, consider that $\Upsilon(u,v) = \Upsilon(u_0, v_0) + \left(-\frac{\xi_v^2}{\xi_u^2}(u - u_0) + \frac{2\xi_v}{\xi_u}(v - v_0)\right)$ for some point $(\xi_u, \xi_v)$ on the line segment $\{(u_0 + \lambda(u - u_0), v_0 + \lambda(v - v_0)) \in \mathbb{R}^2 : 0 \leq \lambda \leq 1\}$. The claimed result follows immediately. $\square$

Proposition 5.1 yields

$$\left|\widehat{\mathcal{I}}_t - \mathcal{I}_t\right| = \left|\int_{-\infty}^{\infty} \Upsilon(\widehat{p}^{\,\varepsilon}(x,t), \widehat{\partial_x p}^{\,\varepsilon}(x,t)) - \Upsilon(p(x,t), \partial_x p(x,t)) \, dx\right| \lesssim \int_{-\infty}^{\infty} \mathcal{R}_1(x,t) + \mathcal{R}_2(x,t) \, dx \tag{25}$$

where

$$\mathcal{R}_1(x,t) = \frac{(|\widehat{\partial_x p}^{\,\varepsilon}(x,t)| \vee |\partial_x p(x,t)|)^2}{(|\widehat{p}^{\,\varepsilon}(x,t)| \wedge |p(x,t)|)^2} |\widehat{p}^{\,\varepsilon}(x,t) - p(x,t)|, \tag{26}$$

$$\mathcal{R}_2(x,t) = \frac{|\widehat{\partial_x p}^{\,\varepsilon}(x,t)| \vee |\partial_x p(x,t)|}{|\widehat{p}^{\,\varepsilon}(x,t)| \wedge |p(x,t)|} |\widehat{\partial_x p}^{\,\varepsilon}(x,t) - \partial_x p(x,t)|. \tag{27}$$

To prove Theorems 2.1 and 2.2, we will focus on bounding (25) in expectation. Theorem 2.3 involves a slightly different estimator, but though the term of focus is different in Section A.3, the essence of the argument still involves the Taylor expansion given by Proposition 5.1.

### A.1. Proof of Theorem 2.1

Recall $\widehat{p}^{\,\varepsilon}$ and $\widehat{\partial_x p}^{\,\varepsilon}$ are given by (10) and (11) respectively in Theorem 2.1. Examining the terms (26) and (27), it is clear that error bounds of these estimators will be needed; Proposition A.1 delivers suitable bounds, and its proof is deferred to Appendix C.

**Proposition A.1.** *If $\alpha \geq 1$, then*

$$E\left(|\widehat{p}^{\,\varepsilon}(x,t) - p(x,t)|^2\right) \lesssim \frac{\varepsilon(x,t)}{n\sqrt{t}} \left(\mathbb{1}_{\{|x| \leq 1\}} + e^{-\frac{(|x|-1)^2}{2t}} \mathbb{1}_{\{|x| > 1\}}\right),$$

$$E\left(\left|\widehat{\partial_x p}^{\,\varepsilon}(x,t) - \partial_x p(x,t)\right|^2\right) \lesssim \frac{\varepsilon(x,t)}{nt^{3/2}},$$

*where $\widehat{p}^{\,\varepsilon}$ and $\widehat{\partial_x p}^{\,\varepsilon}$ are given by (10) and (11) respectively.*

Furthermore, estimates on the size of $p$ and $\partial_x p$ are needed as seen in (26) and (27). The proofs of Lemmas A.2 and A.3 are deferred to Appendix E.

**Lemma A.2.** *If $t < 1$, then*

$$p(x,t) \asymp \begin{cases} 1 & \text{if } |x| \leq 1, \\ \left(1 \wedge \frac{\sqrt{t}}{|x|-1}\right) e^{-\frac{(|x|-1)^2}{2t}} & \text{if } |x| > 1. \end{cases}$$

**Lemma A.3.** *If $\alpha \geq 1$, then*

$$|\partial_x p(x,t)|^2 \lesssim \varphi_t(x+1)^2 + \varphi_t(x-1)^2 + \underline{\varepsilon}(x,t)\left(\mathbb{1}_{\{|x|\leq 1\}} + \frac{e^{-\frac{(|x|-1)^2}{2t}}}{\sqrt{t}}\mathbb{1}_{\{|x|>1\}}\right).$$

With these estimates in hand, we are in position to bound (25) in expectation.

**Proposition A.4.** *If $\alpha \geq 1$ and $t < 1$, then $E\left(\int_{-\infty}^{\infty} \mathcal{R}_1(x,t)\,dx\right) \lesssim \frac{1}{\sqrt{n}t^{3/4}}$, where $\mathcal{R}_1$ is given by (26).*

*Proof.* By $\underline{\varepsilon}(x,t) \leq p(x,t) \leq \overline{\varepsilon}(x,t)$ and Lemmas A.3, E.1, we have

$$E(\mathcal{R}_1(x,t)) \leq \frac{E\left(\left(|\widehat{\partial_x p}^{\varepsilon}(x,t)| \vee |\partial_x p(x,t)|\right)^2 \cdot |\widehat{p}^{\varepsilon}(x,t) - p(x,t)|\right)}{\underline{\varepsilon}(x,t)^2}$$

$$\leq \frac{\varphi_t(x+1)^2 + \varphi_t(x-1)^2 + \underline{\varepsilon}(x,t)\left(\mathbb{1}_{\{|x|\leq 1\}} + \frac{e^{-\frac{(|x|-1)^2}{2t}}}{\sqrt{t}}\mathbb{1}_{\{|x|>1\}}\right)}{\underline{\varepsilon}(x,t)} \cdot \frac{E\left(|\widehat{p}^{\varepsilon}(x,t) - p(x,t)|\right)}{\underline{\varepsilon}(x,t)}. \quad (28)$$

Looking at the first term in the product (28), note $\underline{\varepsilon}(x,t) \asymp 1$ for $|x| \leq 1$ and $\underline{\varepsilon}(x,t) \asymp \left(1 \wedge \frac{\sqrt{t}}{|x|-1}\right)e^{-\frac{(|x|-1)^2}{2t}}$ for $|x| > 1$ as implied by Lemma A.2 since $p(x,t) \asymp \underline{\varepsilon}(x,t)$. Furthermore, $\varphi_t(x+1)^2 \lesssim \frac{1}{\sqrt{t}}\varphi_t(x+1)$ and $\varphi_t(x-1)^2 \lesssim \frac{1}{\sqrt{t}}\varphi_t(x-1)$. Therefore,

$$\frac{\varphi_t(x+1)^2 + \varphi_t(x-1)^2 + \underline{\varepsilon}(x,t)\left(\mathbb{1}_{\{|x|\leq 1\}} + \frac{e^{-\frac{(|x|-1)^2}{2t}}}{\sqrt{t}}\mathbb{1}_{\{|x|>1\}}\right)}{\underline{\varepsilon}(x,t)}$$

$$\lesssim \left(\frac{1}{\sqrt{t}}\left(\varphi_t(x+1) + \varphi_t(x-1)\right) + 1\right)\mathbb{1}_{\{|x|\leq 1\}} + \left(\frac{\varphi_t(x+1)^2 + \varphi_t(x-1)^2}{\underline{\varepsilon}(x,t)} + \frac{e^{-\frac{(|x|-1)^2}{2t}}}{\sqrt{t}}\right)\mathbb{1}_{\{|x|>1\}}$$

$$\lesssim \left(1 + \frac{1}{\sqrt{t}}\left(\varphi_t(x+1) + \varphi_t(x-1)\right)\right)\mathbb{1}_{\{|x|\leq 1\}} + \left(\left(1 \vee \frac{|x|-1}{\sqrt{t}}\right)\cdot\frac{1}{t}e^{-\frac{(|x|-1)^2}{2t}} + \frac{e^{-\frac{(|x|-1)^2}{2t}}}{\sqrt{t}}\right)\mathbb{1}_{\{|x|>1\}}.$$

Let us examine the second term appearing in the product (28). It follows by Jensen's inequality, Proposition A.1, and Lemma A.2 along with $p(x,t) \asymp \underline{\varepsilon}(x,t)$ that

$$\frac{E\left(|\widehat{p}^{\varepsilon}(x,t) - p(x,t)|\right)}{\underline{\varepsilon}(x,t)} \leq \frac{\sqrt{E\left(|\widehat{p}^{\varepsilon}(x,t) - p(x,t)|^2\right)}}{\underline{\varepsilon}(x,t)}$$

$$\leq \frac{\mathbb{1}_{\{|x|\leq 1\}} + e^{-\frac{(|x|-1)^2}{2t}}\mathbb{1}_{\{|x|>1\}}}{\sqrt{n\underline{\varepsilon}(x,t)}t^{1/4}}$$

$$\lesssim \frac{1}{\sqrt{n}t^{1/4}}\mathbb{1}_{\{|x|\leq 1\}} + \frac{1}{\sqrt{n}t^{1/4}}\left(1 \vee \frac{(|x|-1)^{1/2}}{t^{1/4}}\right)e^{-\frac{(|x|-1)^2}{4t}}\mathbb{1}_{\{|x|>1\}}.$$

Therefore,

$$E\left(\int_{-\infty}^{\infty} \mathcal{R}_1(x,t)\,dx\right)$$

$$\lesssim \int_{-\infty}^{\infty}\left(\left(1 + \frac{1}{\sqrt{t}}\left(\varphi_t(x+1) + \varphi_t(x-1)\right)\right)\mathbb{1}_{\{|x|\leq 1\}} + \left(\left(1 \vee \frac{|x|-1}{\sqrt{t}}\right)\cdot\frac{1}{t}e^{-\frac{(|x|-1)^2}{2t}} + \frac{e^{-\frac{(|x|-1)^2}{2t}}}{\sqrt{t}}\right)\mathbb{1}_{\{|x|>1\}}\right)$$

$$\cdot \left(\frac{1}{\sqrt{n}t^{1/4}}\mathbb{1}_{\{|x|\leq 1\}} + \frac{1}{\sqrt{n}t^{1/4}}\left(1 \vee \frac{(|x|-1)^{1/2}}{t^{1/4}}\right)e^{-\frac{(|x|-1)^2}{4t}}\mathbb{1}_{\{|x|>1\}}\right)dx$$

$$= \frac{1}{\sqrt{n}t^{1/4}}\int_{|x|\leq 1} 1 + \frac{\varphi_t(x+1) + \varphi_t(x-1)}{\sqrt{t}}\,dx$$

$$+ \frac{1}{\sqrt{n}t^{3/4}} \int_{|x|>1} \left(1 \vee \frac{|x|-1}{\sqrt{t}}\right) \left(1 \vee \frac{(|x|-1)^{1/2}}{t^{1/4}}\right) \cdot \frac{1}{\sqrt{t}} e^{-\frac{3(|x|-1)^2}{4t}} \, dx$$

$$+ \frac{1}{\sqrt{n}t^{1/4}} \int_{|x|>1} \left(1 \vee \frac{(|x|-1)^{1/2}}{t^{1/4}}\right) \cdot \frac{1}{\sqrt{t}} e^{-\frac{3(|x|-1)^2}{4t}} \, dx$$

$$\lesssim \frac{1}{\sqrt{n}t^{1/4}} \left(1 + \frac{1}{\sqrt{t}}\right) + \frac{1}{\sqrt{n}t^{3/4}} \left(1 + \frac{\sqrt{t}}{\sqrt{t}} + \frac{t^{1/4}}{t^{1/4}}\right) + \frac{1}{\sqrt{n}t^{1/4}} \left(1 + \frac{t^{1/4}}{t^{1/4}}\right)$$

$$\asymp \frac{1}{\sqrt{n}t^{3/4}}.$$

The proof is complete. □

**Proposition A.5.** *If $\alpha \geq 1$ and $t < 1$, then $E\left(\int_{-\infty}^{\infty} \mathcal{R}_2(x,t) \, dx\right) \lesssim \frac{1}{\sqrt{n}t^{3/4}}$, where $\mathcal{R}_2$ is given by (27).*

*Proof.* By $\underline{\varepsilon}(x,t) \leq p(x,t) \leq \overline{\varepsilon}(x,t)$ and Lemmas A.3, E.1, we have

$$E(\mathcal{R}_2(x,t))$$

$$\leq \frac{E\left(|\widehat{\partial_x p}^{\varepsilon}(x,t)| \vee |\partial_x p(x,t)| \cdot |\widehat{\partial_x p}^{\varepsilon}(x,t) - \partial_x p(x,t)|\right)}{\underline{\varepsilon}(x,t)}$$

$$\leq \frac{\varphi_t(x+1) + \varphi_t(x-1) + \sqrt{\underline{\varepsilon}(x,t)}\left(\mathbb{1}_{\{|x|\leq 1\}} + \frac{e^{-\frac{(|x|-1)^2}{4t}}}{t^{1/4}}\mathbb{1}_{\{|x|>1\}}\right)}{\sqrt{\underline{\varepsilon}(x,t)}} \cdot \frac{E\left(|\widehat{\partial_x p}^{\varepsilon}(x,t) - \partial_x p(x,t)|\right)}{\sqrt{\underline{\varepsilon}(x,t)}}. \qquad (29)$$

Looking at the first term in the product (29), note $\underline{\varepsilon}(x,t) \asymp 1$ for $|x| \leq 1$ and $\underline{\varepsilon}(x,t) \asymp \left(1 \wedge \frac{\sqrt{t}}{|x|-1}\right) e^{-\frac{(|x|-1)^2}{2t}}$ for $|x| > 1$ as implied by Lemma A.2 since $p(x,t) \asymp \underline{\varepsilon}(x,t)$. Therefore,

$$\frac{\varphi_t(x+1) + \varphi_t(x-1) + \sqrt{\underline{\varepsilon}(x,t)}\left(\mathbb{1}_{\{|x|\leq 1\}} + \frac{e^{-\frac{(|x|-1)^2}{4t}}}{t^{1/4}}\mathbb{1}_{\{|x|>1\}}\right)}{\sqrt{\underline{\varepsilon}(x,t)}}$$

$$\lesssim (1 + \varphi_t(x+1) + \varphi_t(x-1)) \mathbb{1}_{\{|x|\leq 1\}} + \left(\frac{\varphi_t(x+1) + \varphi_t(x-1)}{\sqrt{\underline{\varepsilon}(x,t)}} + \frac{e^{-\frac{(|x|-1)^2}{4t}}}{t^{1/4}}\right) \mathbb{1}_{\{|x|>1\}}$$

$$\lesssim (1 + \varphi_t(x+1) + \varphi_t(x-1)) \mathbb{1}_{\{|x|\leq 1\}} + \left(\left(1 \vee \frac{(|x|-1)^{1/2}}{t^{1/4}}\right) \cdot \frac{1}{\sqrt{t}} e^{-\frac{(|x|-1)^2}{4t}} + \frac{t^{1/4}}{\sqrt{t}} e^{-\frac{(|x|-1)^2}{2t}}\right) \mathbb{1}_{\{|x|>1\}}.$$

Let us examine the second term appearing in the product (29). It follows by Jensen's inequality and Proposition A.1 that

$$\frac{E\left(|\widehat{\partial_x p}^{\varepsilon}(x,t) - \partial_x p(x,t)|\right)}{\sqrt{\underline{\varepsilon}(x,t)}} \leq \frac{\sqrt{E\left(|\widehat{\partial_x p}^{\varepsilon}(x,t) - \partial_x p(x,t)|^2\right)}}{\sqrt{\underline{\varepsilon}(x,t)}} \leq \frac{1}{\sqrt{n}t^{3/4}}.$$

Therefore,

$$E\left(\int_{-\infty}^{\infty} \mathcal{R}_2(x,t) \, dx\right)$$

$$\lesssim \frac{1}{\sqrt{n}t^{3/4}} \int_{|x|\leq 1} 1 + \varphi_t(x+1) + \varphi_t(x-1) \, dx + \frac{1}{\sqrt{n}t^{3/4}} \int_{|x|>1} \left(1 \vee \frac{(|x|-1)^{1/2}}{t^{1/4}}\right) \cdot \frac{1}{\sqrt{t}} e^{-\frac{(|x|-1)^2}{4t}} + \frac{t^{1/4}}{\sqrt{t}} e^{-\frac{(|x|-1)^2}{2t}} \, dx$$

$$= \frac{1}{\sqrt{n}t^{3/4}} + \frac{1}{\sqrt{n}t^{3/4}} \left(1 + \frac{t^{1/4}}{t^{1/4}} + t^{1/4}\right)$$

$$\asymp \frac{1}{\sqrt{n}t^{3/4}}.$$

The proof is complete. □

Theorem 2.1 follows as a direct consequence.

*Proof of Theorem 2.1.* Proposition 5.1 yields (25). It immediately follows from Propositions A.4 and A.5 that $\sup_{f \in \mathcal{F}_\alpha} E\left(\left|\widehat{\mathcal{I}}_t - \mathcal{I}_t\right|\right) \lesssim \frac{1}{\sqrt{n} t^{3/4}}$, as claimed. $\qquad\square$

### A.2. Proof of Theorem 2.2

In Theorem 2.2, recall $\widehat{p}^\varepsilon$ and $\widehat{\partial_x p}^\varepsilon$ are given by (14) and (15). As they are constructed from preliminary estimators which smooth, an analogous version of Proposition A.1 is needed; Proposition A.6 is such a version, and its proof is deferred to Appendix C.

**Proposition A.6.** *If $\alpha \geq 1$, then*

$$E\left(|\widehat{p}^\varepsilon(x,t) - p(x,t)|^2\right) \lesssim n^{-\frac{2\alpha}{2\alpha+1}} \underline{\varepsilon}(x,t),$$

$$E\left(\left|\widehat{\partial_x p}^\varepsilon(x,t) - \partial_x p(x,t)\right|^2\right) \lesssim n^{-\frac{2(\alpha-1)}{2\alpha+1}} \underline{\varepsilon}(x,t) + (\varphi_t(x+1)^2 + \varphi_t(x-1)^2) n^{-\frac{2\alpha}{2\alpha+1}},$$

*where $\widehat{p}^\varepsilon$ and $\widehat{\partial_x p}^\varepsilon$ are given by (14) and (15) respectively.*

As described in Section A, the proof of Theorem 2.2 proceeds by furnishing a suitable bound for (25) in expectation.

**Proposition A.7.** *If $\alpha \geq 1$ and $t < 1$, then $E\left(\int_{-\infty}^\infty \mathcal{R}_1(x,t)\, dx\right) \lesssim \frac{n^{-\frac{\alpha}{2\alpha+1}}}{\sqrt{t}}$, where $\mathcal{R}_1$ is given by (26).*

*Proof.* By the same calculation leading up to (28), but now using Proposition A.6, we have

$E\left(\mathcal{R}_1(x,t)\right)$

$$\lesssim \frac{\varphi_t(x+1)^2 + \varphi_t(x-1)^2 + \underline{\varepsilon}(x,t)\left(\mathbb{1}_{\{|x|\leq 1\}} + \frac{e^{-\frac{(|x|-1)^2}{2t}}}{\sqrt{t}}\mathbb{1}_{\{|x|>1\}}\right)}{\underline{\varepsilon}(x,t)} \cdot \frac{E\left(|\widehat{p}^\varepsilon(x,t) - p(x,t)|\right)}{\underline{\varepsilon}(x,t)}$$

$$\lesssim \left(\left(\frac{1}{\sqrt{t}}\left(\varphi_t(x+1) + \varphi_t(x-1)\right) + 1\right)\mathbb{1}_{\{|x|\leq 1\}} + \left(\frac{\varphi_t(x+1)^2 + \varphi_t(x-1)^2}{\underline{\varepsilon}(x,t)} + \frac{1}{\sqrt{t}}e^{-\frac{(|x|-1)^2}{2t}}\right)\mathbb{1}_{\{|x|>1\}}\right)$$

$$\cdot \frac{n^{-\frac{\alpha}{2\alpha+1}}}{\sqrt{\underline{\varepsilon}(x,t)}}$$

$$\lesssim \left(\left(\frac{1}{\sqrt{t}}\left(\varphi_t(x+1) + \varphi_t(x-1)\right) + 1\right)\mathbb{1}_{\{|x|\leq 1\}} + \left(\left(1 \vee \frac{|x|-1}{\sqrt{t}}\right)\cdot\frac{1}{t}e^{-\frac{(|x|-1)^2}{2t}} + \frac{1}{\sqrt{t}}e^{-\frac{(|x|-1)^2}{2t}}\right)\mathbb{1}_{\{|x|>1\}}\right)$$

$$\cdot n^{-\frac{\alpha}{2\alpha+1}}\left(\mathbb{1}_{\{|x|\leq 1\}} + \left(1 \vee \frac{(|x|-1)^{1/2}}{t^{1/4}}\right)e^{\frac{(|x|-1)^2}{4t}}\mathbb{1}_{\{|x|>1\}}\right)$$

Integrating over $x$ yields

$$E\left(\int_{-\infty}^\infty \mathcal{R}_1(x,t)\, dx\right) \lesssim n^{-\frac{\alpha}{2\alpha+1}}\left(\frac{1}{\sqrt{t}} + 1\right)$$

$$+ \frac{n^{-\frac{\alpha}{2\alpha+1}}}{\sqrt{t}}\int_{|x|>1}\left(1 \vee \frac{(|x|-1)^{1/2}}{t^{1/4}}\right)\left(1 \vee \frac{|x|-1}{\sqrt{t}}\right)\cdot\frac{1}{\sqrt{t}}e^{-\frac{(|x|-1)^2}{4t}}\, dx$$

$$+ n^{-\frac{\alpha}{2\alpha+1}}\int_{|x|>1}\left(1 \vee \frac{(|x|-1)^{1/2}}{t^{1/4}}\right)\cdot\frac{1}{\sqrt{t}}e^{-\frac{(|x|-1)^2}{4t}}\, dx$$

$$\lesssim \frac{n^{-\frac{\alpha}{2\alpha+1}}}{\sqrt{t}} + \frac{n^{-\frac{\alpha}{2\alpha+1}}}{\sqrt{t}} + n^{-\frac{\alpha}{2\alpha+1}}$$

$$\asymp \frac{n^{-\frac{\alpha}{2\alpha+1}}}{\sqrt{t}}.$$

The proof is complete. $\qquad\square$

**Proposition A.8.** *If $\alpha \geq 1$ and $t < 1$, then $E\left(\int_{-\infty}^{\infty} \mathcal{R}_2(x,t)\,dx\right) \lesssim n^{-\frac{\alpha-1}{2\alpha+1}} + \frac{n^{-\frac{\alpha}{2\alpha+1}}}{\sqrt{t}}$, where $\mathcal{R}_2$ is given by (27).*

*Proof.* By the same calculation leading up to (29), but now using Proposition A.6, we have

$$E\left(\mathcal{R}_2(x,t)\right)$$

$$\lesssim \frac{\varphi_t(x+1) + \varphi_t(x-1) + \sqrt{\underline{\varepsilon}(x,t)}\left(\mathbb{1}_{\{|x|\leq 1\}} + \frac{e^{-\frac{(|x|-1)^2}{4t}}}{t^{1/4}}\mathbb{1}_{\{|x|>1\}}\right)}{\sqrt{\underline{\varepsilon}(x,t)}} \cdot \frac{E\left(\left|\widehat{\partial_x p}^{\varepsilon}(x,t) - \partial_x p(x,t)\right|\right)}{\sqrt{\underline{\varepsilon}(x,t)}}$$

$$\lesssim \left((1 + \varphi_t(x+1) + \varphi_t(x-1))\mathbb{1}_{\{|x|\leq 1\}} + \left(\left(1 \vee \frac{(|x|-1)^{1/2}}{t^{1/4}}\right) \cdot \frac{1}{\sqrt{t}}e^{-\frac{(|x|-1)^2}{4t}} + \frac{t^{1/4}}{\sqrt{t}}e^{-\frac{(|x|-1)^2}{4t}}\right)\mathbb{1}_{\{|x|>1\}}\right)$$

$$\cdot \frac{n^{-\frac{\alpha}{2\alpha+1}}\left(\varphi_t(x+1) + \varphi_t(x-1)\right) + n^{-\frac{(\alpha-1)}{2\alpha+1}}\sqrt{\underline{\varepsilon}(x,t)}}{\sqrt{\underline{\varepsilon}(x,t)}}$$

$$\lesssim \left((1 + \varphi_t(x+1) + \varphi_t(x-1))\mathbb{1}_{\{|x|\leq 1\}} + \left(\left(1 \vee \frac{(|x|-1)^{1/2}}{t^{1/4}}\right) \cdot \frac{1}{\sqrt{t}}e^{-\frac{(|x|-1)^2}{4t}} + \frac{t^{1/4}}{\sqrt{t}}e^{-\frac{(|x|-1)^2}{4t}}\right)\mathbb{1}_{\{|x|>1\}}\right)$$

$$\cdot \left(n^{-\frac{\alpha}{2\alpha+1}}\left((\varphi_t(x+1) + \varphi_t(x-1))\mathbb{1}_{\{|x|\leq 1\}} + \left(1 \vee \frac{(|x|-1)^{1/2}}{t^{1/4}}\right) \cdot \frac{1}{\sqrt{t}}e^{-\frac{(|x|-1)^2}{4t}}\mathbb{1}_{\{|x|>1\}}\right) + n^{-\frac{\alpha-1}{2\alpha+1}}\right)$$

Integrating over $x$ yields

$$E\left(\int_{-\infty}^{\infty} \mathcal{R}_2(x,t)\,dx\right)$$

$$\lesssim n^{-\frac{\alpha-1}{2\alpha+1}} + n^{-\frac{\alpha}{2\alpha+1}} + n^{-\frac{\alpha}{2\alpha+1}}\int_{|x|\leq 1}\varphi_t(x+1)^2 + \varphi_t(x-1)^2\,dx$$

$$+ n^{-\frac{\alpha-1}{2\alpha+1}}\int_{|x|>1}\left(1 \vee \frac{(|x|-1)^{1/2}}{t^{1/4}}\right) \cdot \frac{1}{\sqrt{t}}e^{-\frac{(|x|-1)^2}{4t}} + \frac{t^{1/4}}{\sqrt{t}}e^{-\frac{(|x|-1)^2}{4t}}\,dx$$

$$+ n^{-\frac{\alpha}{2\alpha+1}}\int_{|x|>1}\left(\left(1 \vee \frac{(|x|-1)^{1/2}}{t^{1/4}}\right) \cdot \frac{1}{\sqrt{t}}e^{-\frac{(|x|-1)^2}{4t}}\right)\left(\left(1 \vee \frac{(|x|-1)^{1/2}}{t^{1/4}}\right) \cdot \frac{1}{\sqrt{t}}e^{-\frac{(|x|-1)^2}{4t}} + \frac{t^{1/4}}{\sqrt{t}}e^{-\frac{(|x|-1)^2}{4t}}\right)\,dx$$

$$\lesssim n^{-\frac{\alpha-1}{2\alpha+1}} + \frac{n^{-\frac{\alpha}{2\alpha+1}}}{\sqrt{t}} + n^{-\frac{\alpha-1}{2\alpha+1}}\left(1 + t^{1/4}\right) + \frac{n^{-\frac{\alpha}{2\alpha+1}}}{\sqrt{t}}\left(1 + t^{1/4}\right)$$

$$\asymp n^{-\frac{\alpha-1}{2\alpha+1}} + \frac{n^{-\frac{\alpha}{2\alpha+1}}}{\sqrt{t}}.$$

The proof is complete. $\qquad\square$

*Proof of Theorem 2.2.* It immediately follows from (25) (which is furnished from Proposition 5.1) along with Propositions A.7 and A.8 that $\sup_{f\in\mathcal{F}_\alpha} E\left(\left|\widehat{\mathcal{I}}_t - \mathcal{I}_t\right|\right) \lesssim n^{-\frac{\alpha-1}{2\alpha+1}} + \frac{n^{-\frac{\alpha}{2\alpha+1}}}{\sqrt{t}}.$ $\qquad\square$

### A.3. Proof of Theorem 2.3

To prove Theorem 2.3, the estimation errors of $\widehat{L}_1$ and $\widehat{L}_2$ given by (16) and (17) will be bounded separately. It is immediate to bound the error of $\widehat{L}_1$ since it is unbiased; the bound is stated in Proposition A.9 without proof.

**Proposition A.9.** *We have $E\left(\left|\widehat{L}_1 - L_1\right|\right) \lesssim \frac{1}{\sqrt{n}t^2}$ where $\widehat{L}_1$ is given by (16).*

To bound the estimation error of $\widehat{L}_2$, an argument involving the Taylor expansion of $\Upsilon$ given in Proposition 5.1 will be employed. Proposition 5.1 gives

$$\left|\widehat{L}_2 - L_2\right| = \left|\int_{-\infty}^{\infty}\Upsilon\left(\widehat{p}^{\varepsilon}(x,t), \widehat{q}^{\varepsilon}(x,t)\right) - \Upsilon\left(p^{\varepsilon}(x,t), \frac{1}{t}\int_{-1}^{1}\mu\varphi_t(x-\mu)f(\mu)\,d\mu\right)dx\right|$$

$$\lesssim \int_{-\infty}^{\infty}\mathcal{R}_1(x,t) + \mathcal{R}_2(x,t)\,dx \tag{30}$$

where

$$\mathcal{R}_1(x,t) = \frac{\left(\frac{1}{t}\int_{-1}^{1}\mu\varphi_t(x-\mu)f(\mu)\,d\mu\right)^2 \vee |\widehat{q}^\varepsilon(x,t)|^2}{\left(|\widehat{p}^\varepsilon(x,t)| \wedge |p(x,t)|\right)^2}|\widehat{p}^\varepsilon(x,t) - p(x,t)|, \tag{31}$$

$$\mathcal{R}_2(x,t) = \frac{\left|\frac{1}{t}\int_{-1}^{1}\mu\varphi_t(x-\mu)f(\mu)\,d\mu\right| \vee |\widehat{q}^\varepsilon(x,t)|}{|\widehat{p}^\varepsilon(x,t)| \wedge |p(x,t)|}\left|\widehat{q}^\varepsilon(x,t) - \frac{1}{t}\int_{-1}^{1}\mu\varphi_t(x-\mu)f(\mu)\,d\mu\right|. \tag{32}$$

To bound (30), bounds on the plugged-in estimator's errors are useful. Proposition A.1 already gives a bound on the error of $\widehat{p}^\varepsilon$ given by (10), and the following lemma (whose proof is deferred to Appendix C) provides an error bound for $\hat{q}^\varepsilon$.

**Lemma A.10.** *We have*

$$E\left(\left|\widehat{q}^\varepsilon(x,t) - \frac{1}{t}\int_{-1}^{1}\mu\varphi_t(x-\mu)f(\mu)\,d\mu\right|^2\right) \lesssim \frac{\varepsilon(x,t)}{nt^{5/2}},$$

*where $\widehat{q}^\varepsilon$ is given by (18) and $\underline{\varepsilon}(x,t)$ by (6).*

With these preliminary ingredients in place, (30) can be bounded in expectation.

**Proposition A.11.** *Suppose $c > 0$. If $t \geq c$, then $E\left(\int_{-\infty}^{\infty}\mathcal{R}_1(x,t)\,dx\right) \lesssim \frac{1+c^{-1/2}}{\sqrt{n}t^2}$, where $\mathcal{R}_1$ is given by (31).*

*Proof.* Since $p(x,t) \wedge \hat{p}^\varepsilon(x,t) \gtrsim \underline{\varepsilon}(x,t)$ and $\left|\frac{1}{t}\int_{-1}^{1}\mu\varphi_t(x-\mu)f(\mu)\,d\mu\right| \vee |\tilde{q}^\varepsilon(x,t)| \leq \frac{\overline{\varepsilon}(x,t)}{t} \asymp \frac{\varepsilon(x,t)}{t}$, we have after invoking Proposition A.1, $E(\mathcal{R}_1(x,t)) \lesssim \frac{1}{t^2}E(|\widehat{p}^\varepsilon(x,t) - p(x,t)|) \lesssim \frac{1}{t^2}\cdot\frac{\varepsilon(x,t)^{1/2}}{\sqrt{n}t^{1/4}}$. Integration yields

$$\int_{-\infty}^{\infty}E(\mathcal{R}_1(x,t))\,dx \lesssim \frac{1}{\sqrt{n}t^2}\int_{-\infty}^{\infty}\frac{1}{\sqrt{t}}\left(\int_{-1}^{1}e^{-\frac{(x-\mu)^2}{2t}}\,d\mu\right)^{1/2}dx$$

$$\lesssim \frac{c^{-1/2}}{\sqrt{n}t^2} + \frac{1}{\sqrt{n}t^2}\int_{|x|>1}\frac{1}{\sqrt{t}}\left(\int_{-1}^{1}e^{-\frac{(x-\mu)^2}{2t}}\,d\mu\right)^{1/2}dx$$

$$\lesssim \frac{c^{-1/2}}{\sqrt{n}t^2} + \frac{1}{\sqrt{n}t^2}\int_{|x|>1}\frac{1}{\sqrt{t}}e^{-\frac{(|x|-1)^2}{4t}}\,dx$$

$$\lesssim \frac{c^{-1/2}+1}{\sqrt{n}t^2},$$

as claimed. $\qquad\square$

**Proposition A.12.** *Suppose $c > 0$. If $t \geq c$, then $E\left(\int_{-\infty}^{\infty}\mathcal{R}_2(x,t)\,dx\right) \lesssim \frac{1+c^{-1/2}}{\sqrt{n}t^2}$, where $\mathcal{R}_2$ is given by (32).*

*Proof.* The proof is very similar to the proof of Proposition A.11, except Lemma A.10 is invoked in place of Proposition A.1. By the same reasoning in that proof, we have

$$E(\mathcal{R}_2(x,t)) \lesssim \frac{1}{t}E\left(\left|\widehat{q}^\varepsilon(x,t) - \frac{1}{t}\int_{-1}^{1}\mu\varphi_t(x-\mu)f(\mu)d\mu\right|\right) \lesssim \frac{1}{t}\cdot\frac{\varepsilon(x,t)^{1/2}}{\sqrt{n}t^{5/4}} \lesssim \frac{1}{t^2}\cdot\frac{\varepsilon(x,t)^{1/2}}{\sqrt{n}t^{1/4}}.$$

Integrating and repeating the calculations in the proof of Proposition A.11 yields the desired claim. $\qquad\square$

Theorem 2.3 follows immediately.

*Proof of Theorem 2.3.* It follows from Proposition A.9 that $E\left(\left|\widehat{\mathcal{I}}_t - \mathcal{I}_t\right|\right) \lesssim \frac{1}{\sqrt{n}t^2} + E\left(\left|\widehat{L}_2 - L_2\right|\right)$. Proposition 5.1 yields (30), and (30) is bounded in expectation by Propositions A.11 and A.12, which yields the desired result. $\qquad\square$

## B. Proofs: lower bound

This section contains the proofs of Theorem 3.1 and Theorem 3.2.

## B.1. Proof of Theorem 3.1

The proof of Theorem 3.1 proceeds through Le Cam's two point method applied to $f_0$ and $f_1$ given by (19) and (20) respectively.

**Proposition B.1.** *If $f_0, f_1 \in \mathcal{F}_\alpha$, then*

$$\inf_{\widehat{\mathcal{I}}_t} \sup_{f \in \mathcal{F}_\alpha} E\left(\left|\widehat{\mathcal{I}}_t - \mathcal{I}_t\right|\right) \gtrsim |\mathcal{I}(f_0 * \varphi_t) - \mathcal{I}(f_1 * \varphi_t)| \cdot e^{-n \, d_{KL}(f_1 \| f_0)}.$$

*Proof.* The desired result follows as a straightforward consequence of standard and well-known results (Tsybakov, 2009). $\square$

**Proposition B.2.** *We have $d_{KL}(f_1 \| f_0) \lesssim \epsilon^{2\alpha} \rho$.*

*Proof.* By Lemma 2.7 of (Tsybakov, 2009), it follows $d_{KL}(f_1 \| f_0) \leq \chi^2(f_1 \| f_0) = \int_{-1}^{1} \frac{(f_1(\mu) - f_0(\mu))^2}{f_0(\mu)} d\mu \asymp \int_{-1}^{1} g(\mu)^2 \, d\mu \lesssim \epsilon^{2\alpha} \int_{-\infty}^{\infty} K\left(\frac{\mu}{\rho}\right)^2 d\mu \asymp \epsilon^{2\alpha} \rho$, as claimed. $\square$

To furnish a lower bound on the separation of the Fisher information, namely

$$|\mathcal{I}(f_1 * \varphi_t) - \mathcal{I}(f_0 * \varphi_t)| = \left|\int_{-\infty}^{\infty} \Upsilon(p_1(x,t), \partial_x p_1(x,t)) - \Upsilon(p_0(x,t), \partial_x p_0(x,t)) \, dx\right|,$$

we will use Proposition 5.2. Before doing so, we first present its proof here.

*Proof of Proposition 5.2.* The proof follows from a direct application of Taylor's theorem once the gradient and Hessian are computed. A direct calculation shows $\nabla\Upsilon(u,v) = (-\frac{v^2}{u^2}, \frac{2v}{u})$ and $\nabla^2\Upsilon(u,v) = \begin{pmatrix} \frac{2v^2}{u^3} & -\frac{2v}{u^2} \\ -\frac{2v}{u^2} & \frac{2}{u} \end{pmatrix}$. The proof is complete. $\square$

Recall throughout this section and throughout Appendix D.1, we denote $p_0(x,t) := (f_0 * \varphi_t)(x)$ and $p_1(x,t) := (f_1 * \varphi_t)(x)$. Proposition 5.2 yields the expansion

$$\begin{aligned}
\mathcal{I}(f_1 * \varphi_t) - \mathcal{I}(f_0 * \varphi_t) = &-\int_{-\infty}^{\infty} \frac{(\partial_x p_0(x,t))^2}{p_0(x,t)^2}(p_1(x,t) - p_0(x,t)) \, dx \\
&+ 2\int_{-\infty}^{\infty} \frac{\partial_x p_0(x,t)}{p_0(x,t)}(\partial_x p_1(x,t) - \partial_x p_0(x,t)) \, dx \\
&+ \int_{-\infty}^{\infty} \Gamma_{p_0(x,t), \partial_x p_0(x,t)}(p_1(x,t), \partial_x p_1(x,t)) \, dx.
\end{aligned} \tag{33}$$

Proposition B.3 bounds the remainder term in (33). Its proof is deferred to Appendix D.1.1.

**Proposition B.3.** *If $\alpha \geq 1$ and $0 < t, \rho < 1$, then*

$$\left|\int_{-\infty}^{\infty} \Gamma_{p_0(x,t), \partial_x p_0(x,t)}(p_1(x,t), \partial_x p_1(x,t)) \, dx\right| \lesssim \frac{\epsilon^{2\alpha}}{\sqrt{t}} + \frac{\epsilon^{2\alpha}}{\rho^2}.$$

With the remainder term handled, attention will now be directed towards the main terms of (33). Lemma B.4 simplifies the first term and Lemma B.5 simplifies the second term; their proofs are deferred to Appendix D.1.2.

**Lemma B.4.** *There exists some universal constant $c > 0$ such that for $t < 1$, we have*

$$\left|\left(-\int_{-\infty}^{\infty} \frac{(\partial_x p_0(x,t))^2}{p_0(x,t)^2}(p_1(x,t) - p_0(x,t)) \, dx\right) - \left(-\frac{1}{2}\int_{0}^{\infty} \frac{\varphi_t(x-1)^2}{p_0(x,t)} \cdot \frac{g(x,t)}{p_0(x,t)} \, dx\right)\right| \lesssim \epsilon^\alpha e^{-\frac{c}{t}}$$

*where $g(x,t) := (g * \varphi_t)(x)$.*

**Lemma B.5.** *There exists some universal constant $c > 0$ such that for $t < 1$, we have*

$$\left| 2 \int_{-\infty}^{\infty} \frac{\partial_x p_0(x,t)}{p_0(x,t)} (\partial_x p_1(x,t) - \partial_x p_0(x,t)) \, dx - 2g(1) \int_0^{\infty} \frac{\varphi_t(x-1)^2}{p_0(x,t)} \, dx \right| \lesssim \frac{\epsilon^{\alpha}}{\rho} + \epsilon^{\alpha} e^{-\frac{c}{t}}.$$

Putting together Proposition B.3 with Lemmas B.4 and B.5 delivers the following functional separation.

**Theorem B.6.** *For $\alpha \geq 1$ and $0 < t, \rho < 1$, we have*

$$|\mathcal{I}(f_1 * \varphi_t) - \mathcal{I}(f_0 * \varphi_t)| \geq \frac{c\epsilon^{\alpha}}{\sqrt{t}} - C \left( \frac{\epsilon^{\alpha}}{\rho} + \epsilon^{\alpha} e^{-\frac{c'}{t}} + \frac{\epsilon^{2\alpha}}{\sqrt{t}} + \frac{\epsilon^{2\alpha}}{\rho^2} \right)$$

*for some universal constants $C, c, c' > 0$.*

*Proof.* By Proposition B.3 and Lemmas B.4, B.5, we have

$$\left| (\mathcal{I}(f_1 * \varphi_t) - \mathcal{I}(f_0 * \varphi_t)) - \left( \int_0^{\infty} \frac{\varphi_t(x-1)^2}{p_0(x,t)} \cdot \left( 2g(1) - \frac{g(x,t)}{2p_0(x,t)} \right) dx \right) \right| \leq C \left( \frac{\epsilon^{\alpha}}{\rho} + \epsilon^{\alpha} e^{-\frac{c'}{t}} + \frac{\epsilon^{2\alpha}}{\sqrt{t}} + \frac{\epsilon^{2\alpha}}{\rho^2} \right)$$

for some universal constants $C, c' > 0$. We can recognize the posterior expectation

$$\frac{g(x,t)}{2p_0(x,t)} = \frac{\frac{1}{2} \int_{-1}^1 g(\mu) \varphi_t(x-\mu) \, d\mu}{\frac{1}{2} \int_{-1}^1 \varphi_t(x-\mu) \, d\mu} = E(g(\mu) \mid X_t = x)$$

from the Bayes model $\mu \sim f_0$ and $X_t \mid \mu \sim N(\mu, t)$. Since $g(\mu) \leq g(1)$ (because $K$ is maximized at the origin), it follows that $g(1) - \frac{g(x,t)}{2p_0(x,t)} \geq 0$. Therefore,

$$\int_0^{\infty} \frac{\varphi_t(x-1)^2}{p_0(x,t)} \cdot \left( 2g(1) - \frac{g(x,t)}{2p_0(x,t)} \right) dx \geq \int_0^{\infty} \frac{\varphi_t(x-1)^2}{p_0(x,t)} \cdot g(1) \, dx \geq \frac{c\epsilon^{\alpha}}{\sqrt{t}}$$

for some universal constant $c > 0$. The proof is complete. $\qquad\square$

With Theorem B.6 in hand, the stage is set to prove Theorem 3.1.

*Proof of Theorem 3.1.* Let $c > 0$ be a sufficiently small universal constant and suppose $t \leq c$. Set $\rho = C \left( \sqrt{t} \vee \epsilon \right)$ where $C > 0$ is a sufficiently large universal constant and

$$\epsilon = \frac{1}{C^2} \left( \left( \frac{1}{n\sqrt{t}} \right)^{\frac{1}{2\alpha}} \wedge n^{-\frac{1}{2\alpha+1}} \right).$$

Since $C$ is sufficiently large and $\rho \gtrsim \epsilon$, it is straightforward to verify $f_1 \in \mathcal{F}_{\alpha}$. It is also obvious $f_0 \in \mathcal{F}_{\alpha}$. Therefore, Proposition B.1 yields

$$\inf_{\widehat{\mathcal{I}}_t} \sup_{f \in \mathcal{F}_{\alpha}} E \left( \left| \widehat{\mathcal{I}}_t - \mathcal{I}_t \right| \right) \gtrsim |\mathcal{I}(f_0 * \varphi_t) - \mathcal{I}(f_1 * \varphi_t)|.$$

Here, we have used $n \, d_{\mathrm{KL}}(f_1 \| f_0) \lesssim 1$ from $\epsilon^{2\alpha} \rho \leq \frac{C^{-(4\alpha-1)}}{n}$ and Proposition B.2. Theorem B.6 thus yields, for some universal constants $C', c', c'' > 0$,

$$\inf_{\widehat{\mathcal{I}}_t} \sup_{f \in \mathcal{F}_{\alpha}} E \left( \left| \widehat{\mathcal{I}}_t - \mathcal{I}_t \right| \right) \geq \frac{c'\epsilon^{\alpha}}{\sqrt{t}} - C' \left( \frac{\epsilon^{\alpha}}{\rho} + \epsilon^{\alpha} e^{-\frac{c''}{t}} + \frac{\epsilon^{2\alpha}}{\sqrt{t}} + \frac{\epsilon^{2\alpha}}{\rho^2} \right).$$

By taking $C > 0$ to be larger than a sufficiently large universal constant, we can ensure $\frac{c'\epsilon^{\alpha}}{8\sqrt{t}} \geq \frac{C'\epsilon^{2\alpha}}{\sqrt{t}}$. Furthermore, consider $C' \frac{\epsilon^{2\alpha}}{\rho^2} \leq \frac{C'}{C} \frac{\epsilon^{2\alpha}}{\sqrt{t}\rho} \leq \frac{8C'}{Cc'} \frac{\epsilon^{\alpha}}{\rho} \cdot \frac{c'\epsilon^{\alpha}}{8\sqrt{t}} \leq \frac{8C'}{Cc'} \epsilon^{\alpha-1} \cdot \frac{c'\epsilon^{\alpha}}{8\sqrt{t}} \leq \frac{c'\epsilon^{\alpha}}{8\sqrt{t}}$ where the last inequality follows from $\alpha \geq 1$ and $C > 0$ being chosen sufficiently large. Likewise, taking $C > 0$ sufficiently large gives $\frac{C'\epsilon^{\alpha}}{\rho} \leq \frac{c'\epsilon^{\alpha}}{8\sqrt{t}}$. Furthermore, since $t \leq c$ and $c$ is sufficiently small, we have $\frac{c'\epsilon^{\alpha}}{8\sqrt{t}} \geq C'\epsilon^{\alpha} e^{-\frac{c''}{t}}$. Therefore, it follows

$$\inf_{\widehat{\mathcal{I}}_t} \sup_{f \in \mathcal{F}_{\alpha}} E \left( \left| \widehat{\mathcal{I}}_t - \mathcal{I}_t \right| \right) \geq \frac{c'\epsilon^{\alpha}}{\sqrt{t}} - 4 \cdot \frac{c'\epsilon^{\alpha}}{8\sqrt{t}} \asymp \frac{\epsilon^{\alpha}}{\sqrt{t}} \asymp \frac{1}{\sqrt{n}t^{3/4}} \wedge \frac{n^{-\frac{\alpha}{2\alpha+1}}}{\sqrt{t}},$$

as claimed. $\qquad\square$

### B.2. Proof of Theorem 3.2

Le Cam's method (stated in Proposition B.7 without proof) is applied to $f_0$ and $f_2$ given by (19) and (22) respectively. Denote $p_2(x, t) = (f_2 * \varphi_t)(x)$ throughout this section.

**Proposition B.7.** *If* $f_0, f_2 \in \mathcal{F}_\alpha$, *then*

$$\inf_{\widehat{\mathcal{I}}_t} \sup_{f \in \mathcal{F}_\alpha} E\left(\left|\widehat{\mathcal{I}}_t - \mathcal{I}_t\right|\right) \gtrsim |\mathcal{I}(f_0 * \varphi_t) - \mathcal{I}(f_2 * \varphi_t)| \cdot e^{-n \, \mathrm{d}_{\mathrm{KL}}(f_2 \,||\, f_0)}.$$

**Proposition B.8.** *We have* $\mathrm{d}_{\mathrm{KL}}(f_2 \,||\, f_0) \lesssim \epsilon^{2\alpha} \rho$.

*Proof.* The proof is the same as the proof of Proposition B.2. □

To bound the Fisher information separation $|\mathcal{I}(f_2 * \varphi_t) - \mathcal{I}(f_0 * \varphi_t)|$, a Taylor expansion reminiscent of the proof of Theorem 3.1 will be employed after some simplification. Since $\partial_x p_2(x, t) = -\frac{x}{t} p_2(x, t) + \frac{1}{t} \int_{-1}^{1} \mu \varphi_t(x - \mu) f_2(\mu) \, d\mu$, we have

$$\mathcal{I}(f_2 * \varphi_t) = \int_{-\infty}^{\infty} \frac{(\partial_x p_2(x, t))^2}{p_2(x, t)} \, dx$$

$$= \int_{-\infty}^{\infty} \frac{x^2}{t^2} p_2(x, t) \, dx - \frac{2}{t^2} \int_{-\infty}^{\infty} \int_{-1}^{1} x \mu \varphi_t(x - \mu) f_2(\mu) \, d\mu \, dx + \frac{1}{t^2} \int_{-\infty}^{\infty} \frac{\left(\int_{-1}^{1} \mu \varphi_t(x - \mu) f_2(\mu) \, d\mu\right)^2}{p_2(x, t)} \, dx$$

$$= \int_{-1}^{1} \frac{\mu^2 + t}{t^2} f_2(\mu) \, d\mu - \frac{2}{t^2} \int_{-1}^{1} \mu^2 f_2(\mu) \, d\mu + \frac{1}{t^2} \int_{-\infty}^{\infty} \frac{\left(\int_{-1}^{1} \mu \varphi_t(x - \mu) f_2(\mu) \, d\mu\right)^2}{p_2(x, t)} \, dx$$

$$= \frac{1}{t} - \frac{1}{t^2} \int_{-1}^{1} \mu^2 f_2(\mu) \, d\mu + \frac{1}{t^2} \int_{-\infty}^{\infty} \frac{\left(\int_{-1}^{1} \mu \varphi_t(x - \mu) f_2(\mu) \, d\mu\right)^2}{p_2(x, t)} \, dx.$$

The same expression holds for $\mathcal{I}(f_0 * \varphi_t)$ with $f_0$ and $p_0$ in place of $f_2$ and $p_2$. Therefore,

$$\mathcal{I}(f_2 * \varphi_t) - \mathcal{I}(f_0 * \varphi_t) = \frac{1}{t^2} \int_{-1}^{1} \mu^2 (f_0(\mu) - f_2(\mu)) \, d\mu + \frac{1}{t^2} \int_{-\infty}^{\infty} \Upsilon(p_2(x, t), N_2(x, t)) - \Upsilon(p_0(x, t), N_0(x, t)) \, dx,$$

where $\Upsilon$ is given by (5), $N_2(x, t) = \int_{-1}^{1} \mu \varphi_t(x - \mu) f_2(\mu) \, d\mu$, and $N_0(x, t) = \int_{-1}^{1} \mu \varphi_t(x - \mu) f_0(\mu) \, d\mu$. Taylor expansion (Proposition 5.1) yields

$$\mathcal{I}(f_2 * \varphi_t) - \mathcal{I}(f_0 * \varphi_t) = \frac{1}{t^2} \int_{-1}^{1} \mu^2 (f_0(\mu) - f_2(\mu)) \, d\mu + \frac{1}{t^2} \int_{-\infty}^{\infty} \Gamma_{p_0(x,t), N_0(x,t)}(p_2(x, t), N_2(x, t)) \, dx$$

$$= -\frac{\epsilon^\alpha}{t^2} \int_{-1}^{1} \mu^2 \psi\left(\frac{\mu}{\rho}\right) d\mu + \frac{1}{t^2} \int_{-\infty}^{\infty} \Gamma_{p_0(x,t), N_0(x,t)}(p_2(x, t), N_2(x, t)) \, dx$$

$$= -\frac{\epsilon^\alpha \rho^3}{t^2} \left(\int_{-\infty}^{\infty} \nu^2 \psi(\nu) \, d\nu\right) + \frac{1}{t^2} \int_{-\infty}^{\infty} \Gamma_{p_0(x,t), N_0(x,t)}(p_2(x, t), N_2(x, t)) \, dx, \qquad (34)$$

where $\frac{1}{t^2} |\Gamma_{p_0(x,t), N_0(x,t)}(p_2(x, t), N_2(x, t))| \leq C \left(\mathcal{R}_1(x, t) + \mathcal{R}_2(x, t)\right)$ with

$$\mathcal{R}_1(x, t) = \frac{1}{t^2} \cdot \frac{N_0(x, t)^2 \vee N_2(x, t)^2}{p_0(x, t)^2 \wedge p_2(x, t)^2} |p_2(x, t) - p_0(x, t)|, \qquad (35)$$

$$\mathcal{R}_2(x, t) = \frac{1}{t^2} \cdot \frac{|N_0(x, t)| \vee |N_2(x, t)|}{p_0(x, t) \wedge p_2(x, t)} |N_2(x, t) - N_0(x, t)|. \qquad (36)$$

The terms (35) and (36) are lower order terms as asserted by Lemmas B.9 and B.10.

**Lemma B.9.** *Suppose* $c > 0$. *If* $t \geq c$, *then there exists* $C$ *depending only on* $c$ *such that* $\int_{-\infty}^{\infty} \mathcal{R}_1(x, t) \, dx \leq \frac{C \epsilon^\alpha \rho^3}{t^3}$ *where* $\mathcal{R}_1$ *is given by (35).*

**Lemma B.10.** *Suppose $c > 0$. If $t \geq c$, then there exists $C$ depending only on on $c$ such that $\int_{-\infty}^{\infty} \mathcal{R}_2(x,t)\,dx \lesssim \frac{C\epsilon^\alpha \rho^3}{t^{5/2}}$ where $\mathcal{R}_2$ is given by (36).*

The proofs of Lemmas B.9 and B.10 are deferred to Appendix D.2. We are now ready to prove Theorem 3.2.

*Proof of Theorem 3.2.* Let $C > 0$ be a sufficiently large universal constant and let $t \geq C$. Set $\rho = \frac{1}{2}$ and $\epsilon = \frac{n^{-\frac{1}{2\alpha}}}{C}$. Since $\rho \gtrsim \epsilon$, it is straightforward to verify $f_2 \in \mathcal{F}_\alpha$. Since $f_0 \in \mathcal{F}_\alpha$ also, it follows by Proposition B.7 that

$$\inf_{\widehat{\mathcal{I}}_t} \sup_{f \in \mathcal{F}_\alpha} E\left(\left|\widehat{\mathcal{I}}_t - \mathcal{I}_t\right|\right) \gtrsim |\mathcal{I}(f_0 * \varphi_t) - \mathcal{I}(f_2 * \varphi_t)|.$$

Here, we have used Proposition B.8 and $\epsilon^{2\alpha}\rho \lesssim \frac{1}{n}$ to obtain $n\,\mathrm{d}_{\mathrm{KL}}(f_2 \,\|\, f_0) \lesssim 1$. Therefore, it follows from (34) along with Lemmas B.9 and B.10 that for some universal constants $C', c' > 0$, we have

$$\inf_{\widehat{\mathcal{I}}_t} \sup_{f \in \mathcal{F}_\alpha} E\left(\left|\widehat{\mathcal{I}}_t - \mathcal{I}_t\right|\right) \geq \frac{\epsilon^\alpha \rho^3}{t^2}\left(c' - \left(\frac{C'}{t} + \frac{C'}{t^{1/2}}\right)\right) \gtrsim \frac{1}{\sqrt{n}t^2}.$$

Here, we have used $t \geq C$ for $C > 0$ a sufficiently large universal constant. We have also used $\left|\int_{-\infty}^{\infty} \psi(\nu)\nu^2\,d\nu\right| \gtrsim 1$. The proof is complete. $\qquad\square$

## C. Technical tools: upper bound

This section contains the proofs of Proposition A.1, Proposition A.6, and Lemma A.10, which give error bounds for the plugged-in preliminary estimators. They deal respectively with the high noise regime, the low noise regime, and the very high noise regime.

*Proof of Proposition A.1.* It is clear truncation only improves the estimation error, and so it is immediate that $E(|\widehat{p}^\varepsilon(x,t) - p(x,t)|^2) \leq E(|\widehat{p}(x,t) - p(x,t)|^2)$ and $E\left(\left|\widehat{\partial_x p}^\varepsilon(x,t) - \partial_x p(x,t)\right|^2\right) \leq E\left(|\partial_x \widehat{p}(x,t) - \partial_x p(x,t)|^2\right)$. Therefore, it suffices to focus on $\widehat{p}$ and $\partial_x \widehat{p}$. Note $\widehat{p}(x,t)$ and $\partial_x \widehat{p}(x,t)$ are unbiased estimators for $p(x,t)$ and $\partial_x p(x,t)$ respectively, so it suffices to compute their variances to establish the desired results. By direct calculation,

$$\mathrm{Var}(\widehat{p}(x,t)) = \frac{1}{n^2}\sum_{i=1}^{n} \mathrm{Var}(\varphi_t(x - \mu_i)) \leq \frac{1}{2\pi t n}\int_{-1}^{1} e^{-\frac{(x-\mu)^2}{t}} f(\mu)\,d\mu \lesssim \frac{1}{n\sqrt{2\pi t}}\int_{-1}^{1} \frac{1}{\sqrt{2\pi t}} e^{-\frac{(x-\mu)^2}{t}}\,d\mu$$

For $|x| \leq 1$, we directly obtain the bound $\mathrm{Var}(\widehat{p}(x,t)) \lesssim \frac{\varepsilon(x,t)}{n\sqrt{t}}$ since $e^{-\frac{(x-\mu)^2}{t}} \leq e^{-\frac{(x-\mu)^2}{2t}}$. Suppose $|x| > 1$. Then

$$\frac{1}{n\sqrt{2\pi t}}\int_{-1}^{1} \frac{1}{\sqrt{2\pi t}} e^{-\frac{(x-\mu)^2}{t}}\,d\mu \lesssim \frac{e^{-\frac{(|x|-1)^2}{2t}}}{n\sqrt{t}}\int_{-1}^{1} \frac{1}{\sqrt{2\pi t}} e^{-\frac{(x-\mu)^2}{2t}}\,d\mu \asymp \frac{e^{-\frac{(|x|-1)^2}{2t}}\varepsilon(x,t)}{n\sqrt{t}},$$

which completes the proof for the claimed bound of $\mathrm{Var}(\widehat{p}(x,t))$.

Next, we examine estimation of the derivative $\partial_x p(x,t)$. Observe

$$\begin{aligned}
\mathrm{Var}(\partial_x \widehat{p}(x,t)) &= \frac{1}{n^2}\sum_{i=1}^{n} \mathrm{Var}(\varphi_t'(x - \mu_i)) \\
&= \frac{1}{n}\int_{-1}^{1} \frac{|x - \mu|^2}{t^2} \cdot \frac{1}{2\pi t} e^{-\frac{(x-\mu)^2}{t}} f(\mu)\,d\mu \\
&\lesssim \frac{1}{nt^{3/2}}\int_{-1}^{1} \frac{|x - \mu|^2}{t} \cdot \frac{1}{\sqrt{t}} e^{-\frac{(x-\mu)^2}{t}}\,d\mu \\
&\lesssim \frac{1}{nt^{3/2}}\int_{-1}^{1} \frac{1}{\sqrt{2\pi t}} e^{-\frac{(x-\mu)^2}{2t}}\,d\mu \\
&\asymp \frac{\varepsilon(x,t)}{nt^{3/2}}.
\end{aligned}$$

Here, we have used the inequality $y \leq e^{y^2/2}$. $\qquad\square$

*Proof of Proposition A.6.* Recall that $\widehat{p}^\varepsilon$ and $\widehat{\partial_x p}^\varepsilon$ given by (14) and (15) are obtained by truncating $\widehat{p}$ and $\widehat{\partial_x p}$ given respectively by (12) and (13). Since the truncation only improves estimation error, it follows by Jensen's inequality that

$$E\left(|\widehat{p}^\varepsilon(x,t) - p(x,t)|^2\right) \leq E\left(|\widehat{p}(x,t) - p(x,t)|^2\right) \leq \int_{-1}^1 |\hat{f}(\mu) - f(\mu)|^2 \varphi_t(x-\mu)\,d\mu \lesssim n^{-\frac{2\alpha}{2\alpha+1}} \underline{\varepsilon}(x,t),$$

as desired.

To prove the remaining claim, observe

$$
\begin{aligned}
&E\left(\left|\widehat{\partial_x p}^\varepsilon(x,t) - \partial_x p(x,t)\right|^2\right) \\
&\lesssim E\left(\left|\widehat{\partial_x p}(x,t) - \partial_x p(x,t)\right|^2\right) \\
&\lesssim |\hat{f}(-1) - f(-1)|^2 \varphi_t(x+1)^2 + |\hat{f}(-1) - f(-1)|^2 \varphi_t(x-1)^2 + \int_{-1}^1 |\hat{f}'(\mu) - f'(\mu)|^2 \varphi_t(x-\mu)\,d\mu \\
&\lesssim n^{-\frac{2\alpha}{2\alpha+1}}\left(\varphi_t(x+1)^2 + \varphi_t(x-1)^2\right) + n^{-\frac{2(\alpha-1)}{2\alpha+1}} \underline{\varepsilon}(x,t),
\end{aligned}
$$

where we have used Jensen's inequality to obtain the penultimate line. The proof is complete. $\qquad\square$

*Proof of Lemma A.10.* Since $\left|\int_{-1}^1 \mu\varphi_t(x-\mu)f(\mu)\,d\mu\right| \leq \overline{\varepsilon}(x,t)$ and since truncation only improves the estimation error, it follows

$$
\begin{aligned}
E\left(\left|\widehat{q}^\varepsilon(x,t) - \frac{1}{t}\int_{-1}^1 \mu\varphi_t(x-\mu)f(\mu)\,d\mu\right|^2\right) &\leq E\left(\left|\frac{1}{nt}\sum_{i=1}^n \mu_i\varphi_t(x-\mu_i) - \frac{1}{t}\int_{-1}^1 \mu\varphi_t(x-\mu)f(\mu)\,d\mu\right|^2\right) \\
&= \mathrm{Var}\left(\frac{1}{nt}\sum_{i=1}^n \mu_i\varphi_t(x-\mu_i)\right) \\
&\leq \frac{1}{nt^2}E(\mu_1^2\varphi_t(x-\mu_1)^2) \\
&\lesssim \frac{1}{nt^{5/2}}\int_{-1}^1 \frac{1}{\sqrt{2\pi t}}e^{-\frac{|x-\mu|^2}{t}}\,d\mu \\
&\lesssim \frac{\varepsilon(x,t)}{nt^{5/2}},
\end{aligned}
$$

as claimed. $\qquad\square$

## D. Technical tools: lower bound

Appendix D.1 contains the deferred proofs of results stated in the main text as well as technical lemmas used in the proof of Theorem 3.1. Likewise, the deferred material for the minimax lower bound in the very high noise regime (Theorem 3.2) is contained in Appendix D.2.

### D.1. High noise and low noise regimes

This section contains a useful auxiliary lemma stating some properties of $f_0(\mu) = \frac{1}{2}\mathbb{1}_{\{|\mu|\leq 1\}}$ and $p_0(x,t) := (f_0 * \varphi_t)(x)$. Appendices D.1.1 and D.1.2 respectively contain the deferred proofs for bounding the remainder term in the Taylor expansion of Proposition 5.2 and bounding the functional separation.

**Lemma D.1.** *Let $f_0(\mu) = \frac{1}{2}\mathbb{1}_{\{|\mu|\leq 1\}}$ and $p_0(x,t) = (f_0 * \varphi_t)(x)$. We have*

$$p_0(x,t) = \frac{1}{2}P\{|N(x,t)| \leq 1\},$$

$$\partial_x p_0(x,t) = \frac{\varphi_t(x+1) - \varphi_t(x-1)}{2},$$

$$\partial_{xx} p_0(x,t) = \frac{-\frac{x+1}{t}\varphi_t(x+1) + \frac{x-1}{t}\varphi_t(x-1)}{2}.$$

*Proof.* The definition directly yields $p_0(x,t) = \int_{-1}^{1} f_0(\mu)\varphi_t(x-\mu)\,d\mu = \frac{1}{2}P\{|N(x,t)| \leq 1\}$. The second claim is proved through integration by parts, $\partial_x p_0(x,t) = (f_0 * \varphi_t')(x) = f_0(-1)\varphi_t(x+1) - f_0(1)\varphi_t(x-1) + \int_{-1}^{1} f_0'(\mu)\varphi_t(x-\mu)\,d\mu = \frac{1}{2}(\varphi_t(x+1) - \varphi_t(x-1))$. With the second claim in hand, direct differentiation yields the third claim. □

### D.1.1. PROOF OF PROPOSITION B.3

In this section, we prove Proposition B.3. From Proposition 5.2, we have

$$|\Gamma_{p_0(x,t),\partial_x p_0(x,t)}(p_1(x,t), \partial_x p_1(x,t))| \leq C(\mathcal{R}_1(x,t) + \mathcal{R}_2(x,t) + \mathcal{R}_3(x,t)) \tag{37}$$

where

$$\mathcal{R}_1(x,t) = \frac{|\partial_x p_0(x,t)|^2 \vee |\partial_x p_1(x,t)|^2}{|p_0(x,t)|^3 \wedge |p_1(x,t)|^3}|p_1(x,t) - p_0(x,t)|^2, \tag{38}$$

$$\mathcal{R}_2(x,t) = \frac{1}{|p_0(x,t)| \wedge |p_1(x,t)|}|\partial_x p_1(x,t) - \partial_x p_0(x,t)|^2, \tag{39}$$

$$\mathcal{R}_3(x,t) = \frac{|\partial_x p_0(x,t)| \vee |\partial_x p_1(x,t)|}{|p_0(x,t)|^2 \wedge |p_1(x,t)|^2}|p_1(x,t) - p_0(x,t)||\partial_x p_1(x,t) - \partial_x p_0(x,t)|. \tag{40}$$

In order to prove Proposition B.3, it suffices to furnish suitable bounds on $\mathcal{R}_1, \mathcal{R}_2$, and $\mathcal{R}_3$. The following preliminary result is a crucial ingredient.

**Lemma D.2.** *We have*

$$|p_1(x,t) - p_0(x,t)|^2 \lesssim \epsilon^{2\alpha}\underline{\varepsilon}(x,t)^2,$$

$$|\partial_x p_1(x,t) - \partial_x p_0(x,t)|^2 \lesssim \epsilon^{2\alpha}\left(\varphi_t(x+1)^2 + \varphi_t(x-1)^2 + \frac{\underline{\varepsilon}(x,t)^2}{\rho^2}\right).$$

*Proof.* By direct examination and that $K$ is bounded by a universal constant, we have

$$|p_1(x,t) - p_0(x,t)|^2 = |(g * \varphi_t)(x)|^2$$

$$\lesssim \epsilon^{2\alpha}\left|\int_{-1}^{1} K\left(\frac{\mu}{\rho}\right)\varphi_t(x-\mu)\,d\mu\right|^2 + \epsilon^{2\alpha}\left|\int_{-1}^{1} K\left(\frac{\mu-1}{\rho}\right)\varphi_t(x-\mu)\,d\mu\right|^2$$

$$+ \epsilon^{2\alpha}\left|\int_{-1}^{1} K\left(\frac{\mu+1}{\rho}\right)\varphi_t(x-\mu)\,d\mu\right|^2$$

$$\lesssim \varepsilon^{2\alpha}\underline{\varepsilon}(x,t)^2.$$

To prove the second claim, observe that we can apply integration by parts to obtain

$$|\partial_x p_1(x,t) - \partial_x p_0(x,t)|^2 = |(g * \varphi_t)'(x)|^2$$

$$= \left| g(-1)\varphi_t(x+1) - g(1)\varphi_t(x-1) + \int_{-1}^{1} g'(\mu)\varphi_t(x-\mu)\,d\mu \right|^2$$

$$\lesssim |g(-1)|^2 \varphi_t(x+1)^2 + |g(1)|^2 \varphi_t(x-1)^2 + \left| \int_{-1}^{1} g'(\mu)\varphi_t(x-\mu)\,d\mu \right|^2$$

$$\lesssim \epsilon^{2\alpha} \left( \varphi_t(x+1)^2 + \varphi_t(x-1)^2 + \frac{\varepsilon(x,t)^2}{\rho^2} \right)$$

Here, we have used that $K'$ is bounded by a universal constant. The proof is complete. $\qquad\square$

We can now bound the three terms $\mathcal{R}_1, \mathcal{R}_2$, and $\mathcal{R}_3$.

**Lemma D.3.** *If $\alpha \geq 1$ and $t < 1$, then $\int_{-\infty}^{\infty} \mathcal{R}_1(x,t)\,dx \lesssim \frac{\epsilon^{2\alpha}}{\sqrt{t}}$ where $\mathcal{R}_1$ is given by (38).*

*Proof.* Let us examine the cases $|x| \leq 1$ and $|x| > 1$ separately. Suppose $|x| \leq 1$. Lemmas A.3 and D.2 then yield

$$\mathcal{R}_1(x,t) \lesssim \frac{\varphi_t(x+1)^2 + \varphi_t(x-1)^2 + \varepsilon(x,t)}{\varepsilon(x,t)^3} \cdot \epsilon^{2\alpha}\varepsilon(x,t)^2 \lesssim \frac{\epsilon^{2\alpha}}{\sqrt{t}} \left( \varphi_t(x+1) + \varphi_t(x-1) \right) + \epsilon^{2\alpha}.$$

Here, we have used $\varepsilon(x,t) \asymp 1$ for $|x| < 1$ since $t < 1$, as implied by Lemma A.2. We have also used $\varphi_t(x-1)^2 \lesssim \frac{1}{\sqrt{t}}\varphi_t(x-1)$ and $\varphi_t(x+1)^2 \lesssim \frac{1}{\sqrt{t}}\varphi_t(x+1)$. Now suppose $|x| > 1$. Then

$$\mathcal{R}_1(x,t) \lesssim \epsilon^{2\alpha} \cdot \frac{\varphi_t(x+1)^2 + \varphi_t(x-1)^2}{\varepsilon(x,t)} + \frac{\epsilon^{2\alpha}}{\sqrt{t}} e^{-\frac{(|x|-1)^2}{2t}} \lesssim \epsilon^{2\alpha} \left( 1 \vee \frac{|x|-1}{\sqrt{t}} \right) \frac{1}{t} e^{-\frac{(|x|-1)^2}{2t}} + \frac{\epsilon^{2\alpha}}{\sqrt{t}} e^{-\frac{(|x|-1)^2}{2t}}.$$

Here, we have used $\varepsilon(x,t) \asymp \left( 1 \wedge \frac{\sqrt{t}}{|x|-1} \right) e^{-\frac{(|x|-1)^2}{2t}}$ as implied by Lemma A.2. We have also used $\varphi_t(x-1)^2 + \varphi_t(x+1)^2 \asymp \frac{1}{t} e^{-\frac{(|x|-1)^2}{t}}$.

Putting together the bounds, we obtain $\int_{-\infty}^{\infty} \mathcal{R}_1(x,t)\,dx \lesssim \frac{\epsilon^{2\alpha}}{\sqrt{t}} + \frac{\epsilon^{2\alpha}}{\sqrt{t}} + \epsilon^{2\alpha} \asymp \frac{\epsilon^{2\alpha}}{\sqrt{t}}$, as desired. $\qquad\square$

**Lemma D.4.** *If $t < 1$, then $\int_{-\infty}^{\infty} \mathcal{R}_2(x,t)\,dx \lesssim \frac{\epsilon^{2\alpha}}{\sqrt{t}} + \frac{\epsilon^{2\alpha}}{\rho^2}$, where $\mathcal{R}_2$ is given by (39).*

*Proof.* As in the proof of Lemma D.3, we analyze the cases $|x| \leq 1$ and $|x| > 1$ separately. Suppose $|x| \leq 1$. Then since $p_0(x,t) \wedge p_1(x,t) \asymp \varepsilon(x,t) \asymp 1$, it follows from Lemma D.2 that

$$\mathcal{R}_2(x,t) \asymp |\partial_x p_1(x,t) - \partial_x p_0(x,t)|^2 \lesssim \frac{\epsilon^{2\alpha}}{\sqrt{t}} \left( \varphi_t(x+1) + \varphi_t(x-1) \right) + \frac{\epsilon^{2\alpha}}{\rho^2}.$$

Here, we have used $\varphi_t(x+1)^2 \lesssim \frac{1}{\sqrt{t}}\varphi_t(x+1)$ and $\varphi_t(x-1)^2 \lesssim \frac{1}{\sqrt{t}}\varphi_t(x-1)$.

Now suppose $|x| > 1$. Then $\varepsilon(x,t) \asymp \left( 1 \wedge \frac{\sqrt{t}}{|x|-1} \right) e^{-\frac{(|x|-1)^2}{2t}}$ as implied by Lemma A.2. Therefore,

$$\mathcal{R}_2(x,t) \lesssim \epsilon^{2\alpha} \cdot \frac{\varphi_t(x+1)^2 + \varphi_t(x-1)^2}{\varepsilon(x,t)} + \frac{\epsilon^{2\alpha}}{\rho^2}\varepsilon(x,t) \lesssim \epsilon^{2\alpha} \left( 1 \vee \frac{|x|-1}{\sqrt{t}} \right) \frac{1}{t} e^{-\frac{(|x|-1)^2}{2t}} + \frac{\epsilon^{2\alpha}}{\rho^2} p_0(x,t)\,dx.$$

Putting together our bounds, we obtain $\int_{-\infty}^{\infty} \mathcal{R}_2(x,t)\,dx \lesssim \frac{\epsilon^{2\alpha}}{\sqrt{t}} + \frac{\epsilon^{2\alpha}}{\rho^2}$, as claimed. $\qquad\square$

**Lemma D.5.** *If $\alpha \geq 1$ and $t < 1$, then $\int_{-\infty}^{\infty} \mathcal{R}_3(x,t)\,dx \lesssim \frac{\epsilon^{2\alpha}}{\sqrt{t}} + \frac{\epsilon^{2\alpha}}{\rho}$, where $\mathcal{R}_3$ is given by (40).*

*Proof.* As in the proofs of Lemmas D.3 and D.4, we separately analyze the cases $|x| \leq 1$ and $|x| > 1$. Suppose $|x| \leq 1$. Then $p_0(x,t) \asymp p_1(x,t) \asymp \underline{\varepsilon}(x,t) \asymp 1$ as implied by Lemma A.2, and so Lemmas A.3 and D.2 yield

$$\mathcal{R}_3(x,t) \lesssim \epsilon^{2\alpha} \left( \varphi_t(x+1) + \varphi_t(x-1) + 1 \right) \left( \varphi_t(x+1) + \varphi_t(x-1) + \frac{1}{\rho} \right)$$

$$\lesssim \left( \frac{\epsilon^{2\alpha}}{\sqrt{t}} + \frac{\epsilon^{2\alpha}}{\rho} + \epsilon^{2\alpha} \right) (\varphi_t(x+1) + \varphi_t(x-1)) + \frac{\epsilon^{2\alpha}}{\rho}.$$

Now suppose $|x| > 1$. Then Lemma A.3 yields

$$\mathcal{R}_3(x,t) \lesssim \epsilon^{2\alpha} \frac{\varphi_t(x+1) + \varphi_t(x-1) + \underline{\varepsilon}(x,t)\frac{1}{\sqrt{t}}e^{-\frac{(|x|-1)^2}{2t}}}{\underline{\varepsilon}(x,t)^2} \cdot \underline{\varepsilon}(x,t) \cdot \left( \varphi_t(x-1) + \varphi_t(x+1) + \frac{\underline{\varepsilon}(x,t)}{\rho} \right)$$

$$\asymp \epsilon^{2\alpha} \frac{\varphi_t(x-1)^2 + \varphi_t(x+1)^2}{\underline{\varepsilon}(x,t)} + \epsilon^{2\alpha} \left( \frac{1}{\sqrt{t}} e^{-\frac{(|x|-1)^2}{2t}} + \frac{1}{\rho} \right) (\varphi_t(x-1) + \varphi_t(x+1))$$

$$+ \epsilon^{2\alpha} \frac{\underline{\varepsilon}(x,t)}{\rho} \cdot \frac{1}{\sqrt{t}} e^{-\frac{(|x|-1)^2}{2t}}$$

$$\lesssim \epsilon^{2\alpha} \left( 1 \vee \frac{|x|-1}{\sqrt{t}} \right) \frac{1}{t} e^{-\frac{(|x|-1)^2}{2t}} + \left( \frac{\epsilon^{2\alpha}}{\sqrt{t}} + \frac{\epsilon^{2\alpha}}{\rho} \right) (\varphi_t(x-1) + \varphi_t(x+1)) + \frac{\epsilon^{2\alpha}}{\rho} \cdot \frac{1}{\sqrt{t}} e^{-\frac{(|x|-1)^2}{2t}}.$$

Here, we have used $\underline{\varepsilon}(x,t) \asymp \left( 1 \wedge \frac{\sqrt{t}}{|x|-1} \right) e^{-\frac{(|x|-1)^2}{2t}}$ as implied by Lemma A.2. The final expression follows from the fact $\underline{\varepsilon}(x,t) \lesssim 1$. Putting together our bounds yields $\int_{-\infty}^{\infty} \mathcal{R}_3(x,t)\,dx \lesssim \frac{\epsilon^{2\alpha}}{\sqrt{t}} + \frac{\epsilon^{2\alpha}}{\rho}$, as claimed. Here, we have used $\rho \lesssim 1$ to obtain $\frac{\epsilon^{2\alpha}}{\rho} \gtrsim \epsilon^{2\alpha}$. □

With these bounds in hand, we are in position to prove Proposition B.3.

*Proof of Proposition B.3.* By Proposition 5.2 along with Lemmas D.3, D.4, and D.5 we have

$$\left| \int_{-\infty}^{\infty} \Gamma_{p_0(x,t),\partial_x p_0(x,t)}(p_1(x,t), \partial_x p_1(x,t))\,dx \right|$$

$$\lesssim \left( \int_{-\infty}^{\infty} \mathcal{R}_1(x,t)\,dx \right) + \left( \int_{-\infty}^{\infty} \mathcal{R}_2(x,t)\,dx \right) + \left( \int_{-\infty}^{\infty} \mathcal{R}_3(x,t)\,dx \right)$$

$$\lesssim \frac{\epsilon^{2\alpha}}{\sqrt{t}} + \frac{\epsilon^{2\alpha}}{\rho^2} + \frac{\epsilon^{2\alpha}}{\rho}$$

$$\asymp \frac{\epsilon^{2\alpha}}{\sqrt{t}} + \frac{\epsilon^{2\alpha}}{\rho^2},$$

since $\rho \lesssim 1$, yielding the claimed result. □

### D.1.2. PROOFS OF LEMMA B.4 AND LEMMA B.5

This section contains the deferred proofs of Lemmas B.4 and B.5, which are used in Theorem B.6 to establish a separation in the Fisher informations.

*Proof of Lemma B.4.* Note since $g$ is an even function, the mapping $x \mapsto g(x,t)$ is an even mapping. Therefore, we have $-\int_{-\infty}^{\infty} \frac{(\partial_x p_0(x,t))^2}{p_0(x,t)^2}(p_1(x,t) - p_0(x,t))\,dx = -2\int_{0}^{\infty} \frac{(\partial_x p_0(x,t))^2}{p_0(x,t)^2}g(x,t)\,dx$. Then, from Lemma D.1, it follows

$$-2\int_{0}^{\infty} \frac{(\partial_x p_0(x,t))^2}{p_0(x,t)^2} g(x,t)\,dx$$

$$= -2\int_{0}^{\infty} \frac{\frac{1}{4}(\varphi_t(x+1) - \varphi_t(x-1))^2}{p_0(x,t)^2} g(x,t)\,dx$$

$$= -\frac{1}{2}\int_0^\infty \frac{\varphi_t(x-1)^2 + \varphi_t(x+1)^2 - 2\varphi_t(x-1)\varphi_t(x+1)}{p_0(x,t)} \cdot \frac{g(x,t)}{p_0(x,t)}\,dx$$

$$= -\frac{1}{2}\int_0^\infty \frac{\varphi_t(x-1)^2}{p_0(x,t)} \cdot \frac{g(x,t)}{p_0(x,t)}\,dx - \frac{1}{2}\int_0^\infty \frac{\varphi_t(x+1)^2}{p_0(x,t)} \cdot \frac{g(x,t)}{p_0(x,t)}\,dx + \int_0^\infty \frac{\varphi_t(x-1)\varphi_t(x+1)}{p_0(x,t)} \cdot \frac{g(x,t)}{p_0(x,t)}\,dx.$$

Since $K$ is bounded, it is clear $|g(x,t)| \lesssim \epsilon^\alpha p_0(x,t)$. Then by Lemma A.2 we have

$$\left| -\frac{1}{2}\int_0^\infty \frac{\varphi_t(x+1)^2}{p_0(x,t)} \cdot \frac{g(x,t)}{p_0(x,t)}\,dx \right| \lesssim \epsilon^\alpha \int_0^\infty \frac{1}{2\pi t} e^{-\frac{(x+1)^2}{t}}\left(\mathbb{1}_{\{|x|\le 1\}} + \left(1\vee \frac{x-1}{\sqrt t}\right)e^{-\frac{(x-1)^2}{2t}}\mathbb{1}_{\{|x|>1\}}\right)dx \lesssim \epsilon^\alpha e^{-\frac{c}{t}}$$

for some universal constant $c > 0$. Here, we have used $t < 1$. The proof is complete. $\qquad\square$

*Proof of Lemma B.5.* Let us abuse notation and write $g(x,t) := (g * \varphi_t)(x)$ as in Lemma B.4. Integration by parts gives $\partial_x g(x,t) = g(-1)\varphi_t(x+1) - g(1)\varphi_t(x-1) + \int_{-1}^1 g'(\mu)\varphi_t(x-\mu)\,d\mu$. Since $x \mapsto \partial_x g(x,t)$ and $x \mapsto \partial_x p_0(x,t)$ are both odd mappings, it follows their product is an even mapping, and so Lemma D.1 gives

$$2\int_{-\infty}^\infty \frac{\partial_x p_0(x,t)}{p_0(x,t)}(\partial_x p_1(x,t) - \partial_x p_0(x,t))\,dx$$

$$= 4\int_0^\infty \frac{\partial_x p_0(x,t)}{p_0(x,t)}\partial_x g(x,t)\,dx$$

$$= 2\int_0^\infty \frac{\varphi_t(x+1) - \varphi_t(x-1)}{p_0(x,t)}\left(g(-1)\varphi_t(x+1) - g(1)\varphi_t(x-1) + \int_{-1}^1 g'(\mu)\varphi_t(x-\mu)\,d\mu\right)dx$$

$$= 2g(1)\int_0^\infty \frac{\varphi_t(x-1)^2}{p_0(x,t)}\,dx - 2\int_0^\infty \frac{\varphi_t(x-1)}{p_0(x,t)}\left(\int_{-1}^1 g'(\mu)\varphi_t(x-\mu)\,d\mu\right)dx + O\left(\epsilon^\alpha e^{-\frac{c}{t}}\right).$$

Here, we have used $t < 1$ and the reasoning similar to that employed in the proof of Lemma B.4. Since $K'$ is bounded, we have

$$\left|2\int_0^\infty \frac{\varphi_t(x-1)}{p_0(x,t)}\left(\int_{-1}^1 g'(\mu)\varphi_t(x-\mu)\,d\mu\right)dx\right| \lesssim \frac{\epsilon^\alpha}{\rho}\int_0^\infty \frac{\varphi_t(x-1)}{p_0(x,t)}P\{|N(x,t)| \le 1\}\,dx$$

$$\asymp \frac{\epsilon^\alpha}{\rho}\int_0^\infty \frac{\varphi_t(x-1)}{p_0(x,t)}p_0(x,t)\,dx$$

$$\lesssim \frac{\epsilon^\alpha}{\rho},$$

completing the proof. $\qquad\square$

### D.2. Very high noise regime

This section contains the proofs of Lemmas B.9 and B.10, which are used to show the remainder term in the Taylor expansion (34) is negligible.

*Proof of Lemma B.9.* Since $f_2(\mu) \asymp f_0(\mu) \asymp 1$ for $|\mu| \le 1$ because $f_0, f_2 \in \mathcal{F}_\alpha$, it follows by Jensen's inequality $|N_0(x,t)|^2 \vee |N_2(x,t)|^2 \lesssim \left(\int_{-1}^1 |\mu|\varphi_t(x-\mu)\,d\mu\right)^2$. Likewise, we have $p_2(x,t)\wedge p_0(x,t) \asymp \int_{-1}^1 \varphi_t(x-\mu)\,d\mu$. Therefore, $\frac{|N_0(x,t)|^2\vee|N_1(x,t)|^2}{p_0(x,t)^2\wedge p_0(x,t)^2} \lesssim 1$. Therefore, it follows from the power series expansion $\varphi_t(x-\mu) = \sum_{k=0}^\infty (-1)^k \frac{\varphi_t^{(k)}(x)}{k!}\mu^k$ that

$$\int_{-\infty}^\infty \mathcal{R}_1(x,t)\,dx \lesssim \frac{1}{t^2}\int_{-\infty}^\infty |p_2(x,t) - p_0(x,t)|\,dx$$

$$= \frac{\epsilon^\alpha}{t^2}\int_{-\infty}^\infty \left|\int_{-1}^1 \psi\left(\frac{\mu}{\rho}\right)\varphi_t(x-\mu)\,d\mu\right|dx$$

$$\le \frac{\epsilon^\alpha}{t^2}\sum_{k=0}^\infty \frac{1}{k!}\left|\int_{-1}^1 \psi\left(\frac{\mu}{\rho}\right)\mu^k\,d\mu\right|\int_{-\infty}^\infty |\varphi_t^{(k)}(x)|\,dx$$

$$= \frac{\epsilon^\alpha}{t^2} \sum_{k=2}^\infty \frac{1}{k!} \left| \int_{-1}^1 \psi\left(\frac{\mu}{\rho}\right) \mu^k \, d\mu \right| \left| \int_{-\infty}^\infty |\varphi_t^{(k)}(x)| \, dx \right|.$$

Here, we have used that $\rho < 1$, $\psi$ is supported on $[-1, 1]$, $\int_{-\infty}^\infty \psi(\nu) \, d\nu = 0$, and $\int_{-\infty}^\infty \psi(\nu)\nu \, d\nu = 0$ since $\psi$ is even. Therefore, it follows from Lemma D.6

$$\int_{-\infty}^\infty \mathcal{R}_1(x, t) \, dx \lesssim \frac{\epsilon^\alpha}{t^2} \sum_{k=2}^\infty \frac{\rho^{k+1}}{k! \cdot t^{k/2}} \left| \int_{-\infty}^\infty \psi(\nu)\nu^k \, d\nu \right| \int_{-\infty}^\infty \left| He_k\left(\frac{x}{\sqrt{t}}\right) \right| \varphi_t(x) \, dx$$

$$\lesssim \frac{\epsilon^\alpha}{t^2} \sum_{k=2}^\infty \frac{\rho^{k+1}}{k! \cdot t^{k/2}} \left( \sqrt{2\pi} \cdot k! \right)^{1/2}$$

$$\leq \frac{C\epsilon^\alpha \rho^3}{t^3},$$

for some $C > 0$ depending only on $c$. Here, we have used $\rho < 1$ and $t \geq c$ to obtain the final line. $\qquad\square$

*Proof of Lemma B.10.* The proof is very similar to the proof of Lemma B.9. From that proof, we have $\frac{|N_0(x,t)| \vee |N_1(x,t)|}{p_0(x,t) \wedge p_0(x,t)} \lesssim 1$. Using the power series expansion $\varphi_t(x - \mu) = \sum_{k=0}^\infty (-1)^k \frac{\varphi_t^{(k)}(x)}{k!}$, it follows

$$\int_{-\infty}^\infty \mathcal{R}_2(x, t) \, dx \lesssim \frac{1}{t^2} \int_{-\infty}^\infty |N_2(x, t) - N_0(x, t)| \, dx$$

$$= \frac{\epsilon^\alpha}{t^2} \sum_{k=0}^\infty \frac{1}{k!} \left| \int_{-1}^1 \psi\left(\frac{\mu}{\rho}\right) \mu^{k+1} \right| \left| \int_{-\infty}^\infty |\varphi_t^{(k)}(x)| \, dx \right|$$

$$\lesssim 0 + \frac{\epsilon^\alpha}{t^2} \sum_{k=1}^\infty \frac{\rho^{k+2}}{k! \cdot t^{k/2}} \left| \int_{-1}^1 \psi(\nu)\nu^{k+1} \, d\nu \right| \int_{-\infty}^\infty \left| He_k\left(\frac{x}{\sqrt{t}}\right) \right| \varphi_t(x) \, dx$$

$$\lesssim \frac{\epsilon^\alpha}{t^2} \sum_{k=1}^\infty \frac{\rho^{k+2}}{k! \cdot t^{k/2}} \left( \sqrt{2\pi} \cdot k! \right)^{1/2}$$

$$\lesssim \frac{C\epsilon^\alpha \rho^3}{t^{5/2}},$$

for some $C > 0$ depending only on $c$. As in the proof of Lemma B.9, we have used $\rho < 1$ and $t \geq c$ to obtain the final line. $\qquad\square$

**Lemma D.6.** *Let* $\varphi_t(x) = \frac{1}{\sqrt{2\pi t}} e^{-\frac{x^2}{2t}}$ *denote the probability density function of* $N(0, t)$. *If* $k \geq 1$ *is an integer, then*

$$\varphi_t^{(k)}(x) = \frac{(-1)^k}{t^{k/2}} \cdot He_k\left(\frac{x}{\sqrt{t}}\right) \varphi_t(x),$$

*where* $He_k$ *is the* $k$-*th (probabilist's) Hermite polynomial.*

*Proof.* Let $\varphi(x) := \varphi_1(x)$ denote the density of the standard Gaussian distribution. Recall that $He_k$ is given by $\varphi^{(k)}(x) = (-1)^k He_k(x)\varphi(x)$. Therefore, $\varphi_t^{(k)}(x) = \frac{d^k}{dx^k} \frac{1}{\sqrt{t}} \varphi\left(\frac{x}{\sqrt{t}}\right) = \frac{1}{\sqrt{t}} \cdot \frac{1}{t^{k/2}} \varphi^{(k)}\left(\frac{x}{\sqrt{t}}\right) = \frac{(-1)^k}{t^{k/2}} He_k\left(\frac{x}{\sqrt{t}}\right) \cdot \frac{1}{\sqrt{t}} \varphi\left(\frac{x}{\sqrt{t}}\right) = \frac{(-1)^k}{t^{k/2}} He_k\left(\frac{x}{\sqrt{t}}\right) \varphi_t(x)$ as claimed. $\qquad\square$

## E. Auxiliary results

This section contains some useful assertions as well as deferred proofs of technical results.

*Proof of Lemma A.2.* Since $c_d \leq f \leq C_d$ on its support, it follows

$$p(x, t) = \int_{-1}^1 \varphi_t(x - \mu) f(\mu) \, d\mu \asymp P\{|N(x, t)| \leq 1\}.$$

If $|x| \leq 1$, then clearly $P\{|N(x,t)| \leq 1\} \asymp 1$, and so the claim is proved for this case. The second claim is Lemma 13[2] from (Dou et al., 2024). $\qquad\square$

*Proof of Lemma A.3.* Since $\alpha \geq 1$, it follows $f$ is differentiable on $(-1,1)$ and $|f'(\mu)| \leq C$ for some universal constant $C > 0$. Therefore, integration by parts yields

$$|\partial_x p(x,t)|^2 = \left| f(-1)\varphi_t(x+1) - f(1)\varphi_t(x-1) + \int_{-1}^{1} f'(\mu)\varphi_t(x-\mu)\,d\mu \right|^2$$

$$\lesssim \varphi_t(x+1)^2 + \varphi_t(x-1)^2 + \left| \int_{-1}^{1} f'(\mu)\varphi_t(x-\mu)\,d\mu \right|^2.$$

Suppose $|x| \leq 1$. Then let us apply Jensen's inequality by treating $\varphi_t(x-\mu)$ as a density in $\mu$ to obtain

$$\left| \int_{-1}^{1} f'(\mu)\varphi_t(x-\mu)\,d\mu \right|^2 = \left| \int_{-\infty}^{\infty} f'(\mu)\mathbb{1}_{\{|\mu|\leq 1\}}\varphi_t(x-\mu)\,d\mu \right|^2$$

$$\leq \int_{-\infty}^{\infty} |f'(\mu)|^2 \mathbb{1}_{\{|\mu|\leq 1\}}\varphi_t(x-\mu)\,d\mu$$

$$\lesssim \underline{\varepsilon}(x,t),$$

which yields the desired result in this case. Now suppose $|x| > 1$. Let us apply Jensen's inequality in a different way by treating $\mathbb{1}_{\{|\mu|\leq 1\}}$ as a finite measure in $\mu$ to obtain

$$\left| \int_{-1}^{1} f'(\mu)\varphi_t(x-\mu)\,d\mu \right|^2 \leq \int_{-1}^{1} |f'(\mu)|^2 \varphi_t(x-\mu)^2\,d\mu \lesssim \frac{1}{\sqrt{t}} \int_{-1}^{1} \frac{1}{\sqrt{t}} e^{-\frac{(x-\mu)^2}{t}}\,d\mu \lesssim \frac{e^{-\frac{(|x|-1)^2}{2t}}}{\sqrt{t}}\underline{\varepsilon}(x,t),$$

as claimed. The proof is complete. $\qquad\square$

**Lemma E.1.** *With probability one,* $\left| \widehat{\partial_x p}^\varepsilon(x,t) \right|^2 \lesssim \varphi_t(x+1)^2 + \varphi_t(x-1)^2 + \underline{\varepsilon}(x,t)$ *where* $\widehat{\partial_x p}^\varepsilon$ *is given by (11).*

*Proof.* The proof is immediate by the definition of $\widehat{\partial_x p}^\varepsilon$. $\qquad\square$

**Lemma E.2.** *If* $\rho \in (0,1)$, *then function $g$ is even and* $\int_{-\infty}^{\infty} g(\mu)\,d\mu = 0$.

*Proof.* Since $K$ is symmetric (i.e. even), it immediately follows $g(\mu) = g(-\mu)$ for all $\mu \in \mathbb{R}$. To show the next claim, note that direct calculation yields

$$\int_{-\infty}^{\infty} g(\mu)\,d\mu = -\epsilon^\alpha \int_{-\infty}^{\infty} K\left(\frac{\mu}{\rho}\right)\mathbb{1}_{\{|\mu|\leq 1\}}\,d\mu + \epsilon^\alpha \int_{-\infty}^{\infty} K\left(\frac{\mu-1}{\rho}\right)\mathbb{1}_{\{|\mu|\leq 1\}}\,d\mu$$

$$+ \epsilon^\alpha \int_{-\infty}^{\infty} K\left(\frac{\mu+1}{\rho}\right)\mathbb{1}_{\{|\mu|\leq 1\}}\,d\mu$$

$$= -\epsilon^\alpha \int_{-\rho}^{\rho} K\left(\frac{\mu}{\rho}\right)d\mu + \epsilon^\alpha \int_{1-\rho}^{1} K\left(\frac{\mu-1}{\rho}\right),d\mu$$

$$+ \epsilon^\alpha \int_{-1}^{-1+\rho} K\left(\frac{\mu+1}{\rho}\right)d\mu$$

$$= -\epsilon^\alpha \rho \int_{-1}^{1} K(\nu)\,d\nu + \epsilon^\alpha \rho \int_{-1}^{0} K(\nu)\,d\nu + \epsilon^\alpha \rho \int_{0}^{1} K(\nu)\,d\nu$$

$$= 0,$$

as claimed. $\qquad\square$

---

[2]Note the statement of Lemma 13 has a typo, but its proof yields the expression stated in Lemma A.2.

# F. Proof of Theorem 4.1

*Proof of Theorem 4.1.* Consider $\mu \sim f$ and $X_t := \mu + \sqrt{t}Z$ with $Z \sim N(0,1)$ drawn independently of $\mu$. Let $\tilde{X}_t := \frac{X_t}{\sqrt{t}}$, that is to say, we have just standardized for convenience to obtain $\tilde{X}_t = \frac{1}{\sqrt{t}}\mu + Z$. Note that the density of $\tilde{\mu}$ is $\tilde{f}$ given by $\tilde{f}(u) := \sqrt{t}f(\sqrt{t}u)$. It follows directly that $t\mathcal{I}(f*\varphi_t) = \mathcal{I}(\tilde{f}*\varphi_1) = 1 - \frac{1}{t}E\left((\mu - E(\mu\,|\,\tilde{X}_t))^2\right)$, where the latter equality is Brown's identity (Brown, 1971). Next, the I-MMSE identity (Guo et al., 2005) asserts $\frac{d}{d(1/t)}I(\tilde{X}_t;\mu) = \frac{1}{2}E((\mu - E(\mu\,|\,\tilde{X}_t))^2)$. Since we have the identity $I(X_t;\mu) = I(\tilde{X}_t;\mu)$ for all $t > 0$, it follows $\frac{d}{dt}I(X_t;\mu) = \left(-\frac{1}{t^2}\right)\frac{d}{d(1/t)}I(\tilde{X}_t;\mu) = -\frac{1}{2t^2}E((\mu - E(\mu\,|\,\tilde{X}_t))^2)$. Therefore, we have $\mathcal{I}_t = \mathcal{I}(f*\varphi_t) = \frac{1}{t}\left(1 - \frac{1}{t}E((\mu - E(\mu\,|\,\tilde{X}_t))^2)\right) = \frac{1}{t}\left(1 + 2t\frac{d}{dt}I(X_t;\mu)\right)$ as claimed. $\square$

# G. Notation

For $a, b \in \mathbb{R}$ the notation $a \lesssim b$ denotes the existence of a universal constant $c > 0$ such that $a \leq cb$. The notation $a \gtrsim b$ is used to denote $b \lesssim a$. Additionally $a \asymp b$ denotes $a \lesssim b$ and $a \gtrsim b$. The symbol $:=$ is frequently used when defining a quantity or object. Furthermore, we frequently use $a \vee b := \max(a, b)$ and $a \wedge b := \min(a, b)$. We generically use the notation $\mathbb{1}_A$ to denote the indicator function for an event $A$. For two probability measures $P$ and $Q$ on a measurable space $(\mathcal{X}, \mathcal{A})$, the total variation distance is defined as $d_{\text{TV}}(P, Q) := \sup_{A \in \mathcal{A}} |P(A) - Q(A)|$. The Kullback-Leibler divergence between $P$ and $Q$ is denoted as $d_{\text{KL}}(P\,\|\,Q) = \int \log\left(dP/dQ\right)\,dP$ if $P$ is absolutely continuous with respect to $Q$ and $d_{\text{KL}}(P\,\|\,Q) = \infty$ otherwise. For a probability measure $P$ with a density $p$, the entropy is denoted $h(P) = -\int p(x)\log p(x)\,dx$. We will frequently abuse notation and use the same symbols with densities or random variables in place of measures. For random variables $U \sim P_U$ and $V \sim P_V$ with joint distribution $(U, V) \sim P_{UV}$, the mutual information is $I(U; V) = d_{\text{KL}}(P_{UV}\,\|\,P_U \otimes P_V)$.

# H. Related work on smoothed Fisher information and a finite-sample analysis of location estimation

The smoothed Fisher information has also been recently shown to play a critical role in a finite-sample analysis of the fundamental statistical task of mean estimation (Gupta et al., 2022; 2023a;b). Gupta et al. (2023b) make the following, insightful observation. Consider the problem of estimating the mean $\theta$ of a density $f$ with variance $\sigma^2$, given $n$ i.i.d. samples $X_1, ..., X_n$. Sample mean is an obvious choice of estimator, and by the central limit theorem, it has the asymptotic distribution $\sqrt{n}(\bar{X} - \theta) \implies N(0, \sigma^2)$. However, in some cases of $f$, the mean $\theta$ might also coincide with the *location*, and some location estimators might outperform the sample mean. For example, consider the Laplace distribution centered at $\theta$ with variance 2, which has density $f(x) \propto e^{-|x-\theta|}$. Note $\theta$ is both the mean and the location (which is also the median in this case). It turns out that the sample median, expressed as $X_{(n/2)}$ in order-statistic notation, beats $\bar{X}$ since $\sqrt{n}(X_{(n/2)} - \theta) \implies N(0, 1)$ whereas $\sqrt{n}(\bar{X} - \theta) \implies N(0, 2)$. The asymptotic variance of sample median is half that of sample mean; in fact, sample median is the maximum likelihood estimator and is thus optimal for estimation of $\theta$ in this example.

More generally, better asymptotic variance than $\sigma^2$ can be achieved in the location estimation problem. The location estimation problem is the problem of estimating the ground truth location parameter $\theta^*$ in the parametric family $\{f(x - \theta)\}_{\theta \in \mathbb{R}}$, where $f$ is some known density. It is classical (Vaart, 1998) that the maximum likelihood estimator is asymptotically normal, centered around $\theta^*$, and with variance given by the reciprocal of the Fisher information of $f$. This variance can be substantially smaller than $\sigma^2$. However, $f$ needs to be known up to translation. Gupta et al. (2023b) ask the intriguing question of whether it is possible, in the case of an *unknown* density $f$ that is symmetric about its mean and, whether it is also possible to attain a Fisher-information like speedup in *finite-samples*.

Gupta et al. (2023b) construct an estimator $\hat{\theta}$ which, with probability at least $1 - \delta$ and with $n \gtrsim \log(1/\delta)$ samples, achieves $|\hat{\theta} - \theta| \leq (1 + \eta)\sqrt{\frac{2\log(2/\delta)}{n\mathcal{I}(f*\varphi_t)}}$ for $t \asymp \sigma^2$ and $\eta = (\log(1/\delta)/n)^{1/13}$. Specifically, their bound involves the smoothed Fisher information of $f$ and asserts a speedup since $\frac{1}{\mathcal{I}(f*\varphi_t)} \leq \sigma^2 + t$. Their results build upon earlier work (Gupta et al., 2022; 2023b) which show that the smoothed Fisher information is a fundamental quantity in a finite-sample analysis of the location estimation problem (where $f$ is assumed known up to translation).

The work of Gupta et al. (2023b) gives a point estimator which enjoys faster convergence rates. The natural problem to

consider next is hypothesis testing, or equivalently, construction of confidence intervals. When $f$ is not known, then the error bound of $\hat{\mu}$ is not computable since $\mathcal{I}(f * \varphi_t)$ is itself not known. Consequently, it is desirable to estimate the smoothed Fisher information $\mathcal{I}(f * \varphi_t)$ to address these subsequent problems.

