# OpenReview forum: "Sharp Optimality of Simple, Plug-in Estimation of the Fisher Information of a Smoothed Density"
_ICML.cc/2025/Conference — ICML 2025 poster_

### Official Review · Reviewer_SWHe · 2025-03-02

**Overall Recommendation:** 3

**Summary:**

This paper analyzes the minimax rate for estimation of the Fisher Information of a 1-dimensional Gaussian-smoothed density that satisfy an alpha-Holder condition from samples. It shows that variants of the simple plug-in estimator achieves the minimax rate, which varies depending on the amount of Gaussian smoothing, and proves matching lower bounds. It also shows the implications of this result on estimation of mutual information and entropy.

**Claims And Evidence:**

Claims are supported by rigorous proof

**Essential References Not Discussed:**

The recent works below analyze mean and location estimation of Gaussian-smoothed densities, which is directly related to the present paper, and so, should be discussed.

1) Finite-Sample Symmetric Mean Estimation with Fisher Information Rate. Shivam Gupta, Jasper C.H. Lee, and Eric Price. COLT 2023
2) High-Dimensional Location Estimation via Norm Concentration for Subgamma Vectors. Shivam Gupta, Jasper C.H. Lee, and Eric Price. ICML 2023
3) Finite-Sample Maximum Likelihood Estimation of Location. Shivam Gupta, Jasper C.H. Lee, Eric Price, and Paul Valiant. NeurIPS 2022

**Experimental Designs Or Analyses:**

No experiments.

**Methods And Evaluation Criteria:**

No empirical results

**Other Comments Or Suggestions:**

N/A

**Other Strengths And Weaknesses:**

It would be nice to include more quantitative intuition for the rates obtained -- currently the lower and upper bounds are explained well qualitatively, but a clear quantitative explanation is lacking.

**Questions For Authors:**

- Is there any clear and concise quantitative explanation that you can provide for the rates obtained?

**Relation To Broader Scientific Literature:**

Several works have looked at estimation of the mean and location of Gaussian smoothed densities with error rate depending on the Fisher Information, which relates directly to the results in this paper (see 1, 2, 3 below). There have been many works looking at density estimation of smoothed densities (for example Goldfeld et al 2020), which are mentioned. There are also previous works on Fisher Information estimation, which are also mentioned.

1) Finite-Sample Symmetric Mean Estimation with Fisher Information Rate. Shivam Gupta, Jasper C.H. Lee, and Eric Price. COLT 2023
2) High-Dimensional Location Estimation via Norm Concentration for Subgamma Vectors. Shivam Gupta, Jasper C.H. Lee, and Eric Price. ICML 2023
3) Finite-Sample Maximum Likelihood Estimation of Location. Shivam Gupta, Jasper C.H. Lee, Eric Price, and Paul Valiant. NeurIPS 2022

**Theoretical Claims:**

Yes, they seem to be correct -- I checked all of them.

---

> ### Author Rebuttal · Authors · 2025-03-29
>
> Thank you for your helpful review!
>
> __Essential references not discussed:__
> Thanks for pointing these very relevant papers out! We plan to add the following text (perhaps with some modification to obey space constraints) to the revised manuscript.
>
> "The smoothed Fisher information has also been recently shown to play a critical role in a finite-sample analysis of the fundamental statistical task of mean estimation (Gupta et al., 2022; Gupta et al., 2023; Gupta et al., 2023). Gupta et al. (2023) make the following, insightful observation. Consider the problem of estimating the mean $\theta$ of a density $f$ with variance $\sigma^2$, given i.i.d. samples $X_1,...,X_n$. Sample mean is an obvious choice of estimator. However, in some cases of $f$, the mean $\theta$ might also coincide with the *location*, and some location estimators might outperform the sample mean. For example, consider the Laplace distribution centered at $\theta$ with variance $2$, which has density $f(x) \propto e^{-|x-\theta|}$. Note $\theta$ is both the mean and the location (which is also the median in this case). It turns out that the sample median, expressed as $X_{(n/2)}$ in order-statistic notation, beats $\bar{X}$ since $\sqrt{n}(X_{(n/2)} - \theta) \implies N(0, 1)$ whereas $\sqrt{n}(\bar{X} - \theta) \implies N(0, 2)$. The asymptotic variance of sample median is half that of sample mean; in fact, sample median is the maximum likelihood estimator and is thus optimal for estimation of $\theta$ in this example.
>
> More generally, better asymptotic variance than $\sigma^2$ can be achieved in the location estimation problem. The location estimation problem is the problem of estimating the ground truth location parameter $\theta^*$ in the parametric family $\{f(x-\theta)\}_{\theta \in \mathbb{R}}$, where $f$ is some known density. It is classical that the maximum likelihood estimator is asymptotically normal, centered around $\theta^*$, and with variance given by the reciprocal of the Fisher information of $f$. This variance can be substantially smaller than $\sigma^2$. Gupta et al. (2023) ask the intriguing question of whether it is possible, in the case of an *unknown* density $f$ that is symmetric about its mean and, whether it is also possible to attain a Fisher-information like speedup in *finite-samples*.
>
> Gupta et al. (2023) construct an estimator $\hat{\theta}$ which, with probability at least $1-\delta$ and with $n \gtrsim \log\left(1/\delta\right)$ samples, achieves $|\hat{\theta} - \theta| \leq (1+\eta)\sqrt{\frac{2\log(2/\delta)}{n \mathcal{I}(f*\varphi\_t)}}$ for $t \asymp \sigma^2$ and $\eta = (\log(1/\delta)/n)^{1/13}$. Specifically, their bound involves the smoothed Fisher information of $f$ and asserts a speedup since $\frac{1}{\mathcal{I}(f*\varphi\_t)} \leq \sigma^2 + t$. Their results build upon earlier work (Gupta et al., 2022; Gupta et al., 2023) which show that the smoothed Fisher information is a fundamental quantity in location estimation.
>
> The work of Gupta et al. (2023) gives a point estimator which enjoys faster convergence rates. The natural problem to consider next is hypothesis testing, or equivalently, construction of confidence intervals. When $f$ is not known, then the error bound of $\hat{\mu}$ is not computable since $\mathcal{I}(f*\varphi\_t)$ is itself not known. Consequently, it is desirable to estimate the smoothed Fisher information $\mathcal{I}(f*\varphi\_t)$ to address these subsequent problems."
>
> __Questions for authors:__
>
> There is some quantitative intuition which can be given. We can look to the estimation rates of the plugged-in targets. Estimation of the derivative is the harder problem, and is thus the rate-dominating step.
>
> In the high noise regime, derivative estimation is done by truncating $\partial_x \hat{p}(x, t) = \frac{1}{n} \sum_{i=1}^{n} \varphi_t'(x-\mu_i)$, and the truncation only improves the estimation error. This is exactly a kernel density estimator for the derivative with kernel $\varphi_t'$ with bandwidth $\sqrt{t}$. However, observe our target is not $f'$ but actually $\partial_x p = f*\varphi_t'$, so no bias is incurred. Hence, the error is given by the variance, which is well known to be $\frac{1}{n\sqrt{t}^3}$ in squared loss, which yields $\frac{1}{\sqrt{n}t^{3/4}}$ in absolute loss.
>
> In the low noise regime, recall we express via integration by parts $\partial_x p(x, t) = (f*\varphi_t)'(x) = f(-1)\varphi_t(x+1) - f(1)\varphi_t(x-1) + \int_{-1}^{1} f'(\mu) \varphi_t(x-\mu) d\mu$. Ignoring the truncation, we estimate by plugging in estimators for the unknown quantities. In Theorem 2.2, the term $n^{-\frac{\alpha-1}{2\alpha+1}}$ comes from plugging in $\hat{f}'$. The term $\frac{n^{-\frac{\alpha}{2\alpha+1}}}{\sqrt{t}}$ comes from plugging in $\hat{f}(1)$ and $\hat{f}(-1)$. The factor $\frac{1}{\sqrt{t}}$ comes from $\varphi_t(x+1) \asymp \frac{1}{\sqrt{t}}$ for $|x+1| \lesssim \sqrt{t}$ (and likewise with $\varphi_t(x-1)$ for $|x-1| \lesssim \sqrt{t}$).

---

### Official Review · Reviewer_CeFJ · 2025-03-06

**Overall Recommendation:** 4

**Summary:**

This paper studies the problem of estimating the Fisher information of smoothed probability densities falling in the $\alpha$-Holder smooth class. The authors derive minimax rate bounds for the plug-in estimator, showing that a simple plug-in estimator is optimal for smoothed probability densities. The convergence results are further extended for mutual information and entropy estimation.

**Claims And Evidence:**

The main claim of the paper: The plug-in estimator is optimal for smoothed probability densities, is well-supported by thorough theoretical analysis.

**Essential References Not Discussed:**

No

**Experimental Designs Or Analyses:**

NA

**Methods And Evaluation Criteria:**

NA

**Other Comments Or Suggestions:**

It would be more convincing if the authors could provide some empirical results to verify their main claim, e.g. showing how the estimation precision changes with $t$ and $\alpha$.

**Other Strengths And Weaknesses:**

Strength:
The presented results are new as far as I know. The main claim is of significant importance to the community to design efficient algorithms for the estimation of information quantities.

Weakness:
There are some restrictions on applicable probability density, e.g. only bounded densities are considered. There is also a gap for $c < t < C$.

**Questions For Authors:**

What will happen if the smoothing kernel is not Gaussian? e.g. $\phi_t(x) \propto \frac{1}{t(x^2 + 1)}$? Do the main results still hold for such kinds of smoothing?

**Relation To Broader Scientific Literature:**

The main result is of significant importance for designing efficient approximations for information quantities including Fisher information, mutual information and entropy. It complements previous results in estimators for unsmoothed probability densities.

**Theoretical Claims:**

I went through the proofs quickly, and they look good to me. I have not checked each detail in the proofs.

---

> ### Author Rebuttal · Authors · 2025-03-29
>
> Thanks for the thoughtful report!
>
> __Other suggestions 1:__ Reviewers pEyQ and arrG also asked about computation, which is related to your comment. Let us first describe how computation of the estimators $\widehat{\mathcal{I}}\_t$ can be done, since it involves an integral over an infinite interval.
>
> We can approximate the integral and still obtain the claimed statistical rate. The idea is simple, we simply truncate the integral to a large enough, bounded interval (whose length is growing in $n$). The error from ignoring the complement of the interval turns out to be negligible. For illustration, let us just discuss the high and low noise regimes, where we plug-in estimators of the density and its derivative. A similar argument will hold for the very high noise regime. Let $R > 0$ be a hyperparameter to be tuned later. We will approximate by
> $$
> \widehat{\mathcal{I}}\_{t, R} := \int_{-R}^{R} \frac{\widehat{\partial_x p}^\varepsilon(x, t)^2}{\hat{p}^\varepsilon(x, t)} dx.
> $$
> We can approximate this integral by Monte Carlo,
> $$
> \widehat{\mathcal{I}}\_{t, R}^{M} := \frac{2R}{M}\sum_{i=1}^{M} \frac{\widehat{\partial_x p}^\varepsilon(X_i, t)^2}{\hat{p}^\varepsilon(X_i, t)}
> $$
> where $\{X\_i\}\_{i=1}^{M}$ are $M$ i.i.d. points drawn uniformly in $[-R, R]$. The estimation error can be bounded as
> $$
> E\left(\left|\widehat{\mathcal{I}}\_{t, R}^{M} - \mathcal{I}\_t\right|\right) \leq E\left(\left|\widehat{\mathcal{I}}\_{t, R}^{M} - \widehat{\mathcal{I}}\_{t, R}\right|\right) + E\left(\left|\widehat{\mathcal{I}}\_{t, R} - \widehat{\mathcal{I}}\_{t}\right|\right) + E\left(\left|\widehat{\mathcal{I}}\_{t} - \mathcal{I}\_{t}\right|\right).
> $$
> The last term is exactly the statistical rate we want. The first term can be made to be of smaller order by taking \(M\) sufficiently large. It remains to argue about the second term. To do so, consider that for $|x| > 1$, we have from calculations similar to those employed frequently in the paper (e.g. using Lemmas A.2 and A.3)
> $$
> \frac{\left|\widehat{\partial_xp}^{\varepsilon}(x, t)\right|^2}{\hat{p}^\varepsilon(x, t)} \leq \frac{\overline{\varepsilon}'(x, t)^2}{\underline{\varepsilon}(x, t)} \lesssim \frac{\varphi_t(x-1)^2 + \varphi_t(x+1)^2 + \underline{\varepsilon}(x, t) \frac{e^{-\frac{(|x|-1)^2}{2t}}}{\sqrt{t}}}{\underline{\varepsilon}(x, t)} \lesssim \left(1 \vee \frac{|x|-1}{\sqrt{t}}\right) \cdot \frac{1}{t} e^{-\frac{(|x|-1)^2}{2t}} + \frac{e^{-\frac{(|x|-1)^2}{2t}}}{\sqrt{t}}.
> $$
> Therefore, if we pick $R \geq 1 + \sqrt{Ct\log(nt)}$ for a large universal constant $C$, we have
> $$
> \left|\widehat{\mathcal{I}}\_{t, R} - \widehat{\mathcal{I}}\_t\right| \lesssim \int_{|x| > R} \left(1 \vee \frac{|x|-1}{\sqrt{t}}\right) \cdot \frac{1}{t} e^{-\frac{(|x|-1)^2}{2t}} + \frac{e^{-\frac{(|x|-1)^2}{2t}}}{\sqrt{t}} dx \leq \frac{1}{(nt)^{\tilde{C}}}
> $$
> for some universal constant $\tilde{C}$ which can be taken to be sufficiently large by taking $C$ sufficiently large. Therefore, the error incurred by approximating $\widehat{\mathcal{I}}\_t$ by $\widehat{\mathcal{I}}\_{t, R}$ is dominated by the desired statistical rate.
>
> The estimators in Section 4 (e.g. Theorem 4.2) can be estimated by the same Monte Carlo strategy. It can also be shown that the error of the complement can be made negligible.
>
> We did a numerical experiment computing our Fisher information estimators on data sampled from the uniform distribution on $[-1, 1]$. Unfortunately, it is not clear to us how to present that figure in our response here, as it seems there is no capability of attaching images in author responses.
>
> __Question for authors 1:__ Thanks for the great question! We imagine perhaps you are thinking about convolving with a Cauchy density instead of a Gaussian density. We imagine it was perhaps meant to be written as $\phi_t(x) \propto \frac{1}{\left(x/\sqrt{t}\right)^2 + 1}$. It's a nice question as Cauchy has no moments.
>
> From a methodological point of view, we feel that the same plug-in strategy can be straightforwardly extended to the Cauchy case. For example, in the high-noise regime it is plausible to truncate $\frac{1}{n}\sum_{i=1}^{n} \phi_t(x-\mu_i)$ and $\frac{1}{n}\sum_{i=1}^{n} \phi_t'(x-\mu_i)$ for use as estimators of $p(x, t)$ and $\partial_x p(x, t)$ respectively (and similar extensions for the other two regimes). To us, the heavier tail behavior does not appear to cause major obstacles. In fact, most of our intuition treats $\phi_t$ as any kernel in a kernel-density estimator, in which case the precise form (Gaussian or Cauchy) doesn't really seem to matter. From the side of rigorous, mathematical analysis, it would seem the arguments would need to be modified to handle the Cauchy case. Our arguments frequently make use of the exponential tail of the Gaussian density to argue various remainder terms can be neglected. It is not clear to us whether serious changes to the broad proof strategy would be needed, or whether just careful, albeit tedious, technical modifications would suffice.

---

> > ### Comment · Reviewer_CeFJ · 2025-04-01
> >
> > Thanks for the clarification about Cauchy kernels. As for the experiments, including a table about some critical points in that figure should suffice for verification. Anyway, it's fine not to include empirical results for strong theoretical papers like this one. I will keep my current score.

---

### Official Review · Reviewer_arrG · 2025-03-11

**Overall Recommendation:** 3

**Summary:**

This paper studies estimation of the Fisher information $\mathcal{I}(f * \psi_t)$ of a smoothed density $\psi_t$, where $\psi_t$ is the Gaussian kernel of bandwidth $t$, given IID samples from a density $f$. Plug-in estimators are proposed, based on appropriately truncated and smoothed estimates of $f * \psi_t$ and its spatial derivative. The paper first presents upper bounds for these estimators, distinguishing between three regimes of the noise magnitude $t$. The paper then presents matching lower bounds for most cases. Finally, using information theoretic identities relating the Fisher information to mutual information and entropy in certain cases, the paper presents and bounds the error of estimators for those latter quantities.

**Claims And Evidence:**

As noted below, I am confused about why Corollary 4.4 holds.

**Essential References Not Discussed:**

N/A

**Experimental Designs Or Analyses:**

The paper does not include any experiments.

**Methods And Evaluation Criteria:**

The paper does not include any experiments.

**Other Comments Or Suggestions:**

1) Typo: Eq. (2): "for all $x, y \in (-1, 1)$" should be "$\mu, \mu' \in (-1, 1)$"

**Other Strengths And Weaknesses:**

There are two main limitations of the proposed estimators that seem make them unusable in practice:
1) The construction of the estimators assumes knowledge of the constants $c_d$, $C_d$, $L$, and $\alpha$. However, these are rarely known in practice. Is it possible to relax this assumption (e.g., by using some surrogates for $c_d$ or $C_d$ in terms of other known quantities) in a way that does not affect the convergence rates? Or is there at least some practical way to get, e.g., the right order of magnitude for these quantities?
2) The mutual information estimator used in Theorem 4.2 requires integrating the Fisher information estimator $\widehat{\mathcal{I}}_s$ over $s$ from $t$ to $\infty$. Can this integral actually by computed? It seems unlikely to me, given the complex dependence of the estimator on $s$, but there might be some tricks that make this possible. Alternatively, is there a computable approximation that can be shown to converge at the claimed rate? Are any additional assumptions needed to show this?
3) Related to the above points, the paper would be made stronger if it demonstrated that the proposed estimators could actually be computed and used in an real-world, or at least simulated, problem.

**Questions For Authors:**

1) The paper only seems to discuss the 1-dimensional case of a density on $\mathbb{R}$; how do results and analysis change for a multi-variate density on $\mathbb{R}^d$?
2) I am confused about why Corollary 4.4 holds. The integral $\int_0^t \mathcal{I}_s ds$ involves estimating the Fisher information in the low-noise regime $s \leq n^{-\frac{2}{2\alpha+1}}$, where Theorem 2.2 gives only a nonparametric convergence rate of order $n^{-\frac{\alpha-1}{2\alpha+1}} + n^{-\frac{\alpha}{2\alpha+1}}/\sqrt{t}$. From this, how can we get the parametric rate $1/\sqrt{n}$? This point has to be clarified for me to accept the paper.

**Relation To Broader Scientific Literature:**

The paper lies in the intersection of classical work on nonparametric estimation of functionals of smooth probability densities and more recent work on estimating densities after Gaussian smoothing.

**Theoretical Claims:**

I did not read the proofs in the supplement, although the high-level descriptions in the main paper generally made sense to me.

---

> ### Author Rebuttal · Authors · 2025-03-29
>
> Thanks for the great feedback!
>
> __Questions for Authors 1:__ Reviewer pEyQ asked the same question; please see our response. Thanks!
>
> __Questions for Authors 2:__ Thank you for the question, and we agree this point could have, and should have, been made clearer in the paper.
>
> As you point out, we estimate the integral $\int\_{0}^{t} \mathcal{I}\_s ds$ by the plug-in $\int\_{0}^{t} \widehat{\mathcal{I}}\_s ds$. Since your question specifically asks about the low-noise regime, let us focus our discussion by examining $t \lesssim n^{-\frac{2}{2\alpha+1}}$. Then for all $s \leq t$, we have from Theorem 2.2 that $E\left(\left|\widehat{\mathcal{I}}_s - \mathcal{I}_s\right|\right) \lesssim n^{-\frac{\alpha-1}{2\alpha+1}} + \frac{n^{-\frac{\alpha}{2\alpha+1}}}{\sqrt{s}}$. Since $s \lesssim n^{-\frac{2}{2\alpha+1}}$, it follows that
>
> $$
> n^{-\frac{\alpha-1}{2\alpha+1}} + \frac{n^{-\frac{\alpha}{2\alpha+1}}}{\sqrt{s}} \asymp n^{-\frac{\alpha}{2\alpha+1}} \cdot n^{\frac{1}{2\alpha+1}} + \frac{n^{-\frac{\alpha}{2\alpha+1}}}{\sqrt{s}} \lesssim n^{-\frac{\alpha}{2\alpha+1}} \cdot s^{-1/2} +  \frac{n^{-\frac{\alpha}{2\alpha+1}}}{\sqrt{s}} \asymp \frac{n^{-\frac{\alpha}{2\alpha+1}}}{\sqrt{s}}.
> $$
>
> Therefore, we have shown $E\left(\left|\widehat{\mathcal{I}}\_s - \mathcal{I}\_s\right|\right) \lesssim \frac{n^{-\frac{\alpha}{2\alpha+1}}}{\sqrt{s}}$. We can now bound the estimation error. Consider,
>
> $$
> E\left(\left|\int\_{0}^{t} \widehat{\mathcal{I}}\_s ds - \int\_{0}^{t} \mathcal{I}\_s ds \right|\right) \leq \int\_{0}^{t} E\left(\left|\widehat{\mathcal{I}}\_s - \mathcal{I}\_s\right|\right) ds \\\\
>             \lesssim \int\_{0}^{t} \frac{n^{-\frac{\alpha}{2\alpha+1}}}{\sqrt{s}}ds \\\\
>             =\left.\left(n^{-\frac{\alpha}{2\alpha+1}}\right) \cdot 2\sqrt{s}\right|\_{s = 0}^{t} \\\\
>             \asymp \sqrt{t} \cdot n^{-\frac{\alpha}{2\alpha+1}}.
> $$
> Since $t \lesssim n^{-\frac{2}{2\alpha+1}}$, we have
> $$
> \sqrt{t} \cdot n^{-\frac{\alpha}{2\alpha+1}} \lesssim n^{-\frac{1}{2\alpha+1}} \cdot n^{-\frac{\alpha}{2\alpha+1}} \asymp n^{-\frac{\alpha+1}{2\alpha+1}}.
> $$
>
> Since $\frac{\alpha+1}{2\alpha+1} \geq \frac{1}{2}$, it follows $n^{-\frac{\alpha+1}{2\alpha+1}} \lesssim \frac{1}{\sqrt{n}}$, and so we have obtained the parametric rate.  Thanks again for your helpful question as these clarifications will improve the paper.
>
> __Other suggestions 1:__ Thanks!
>
> __Weakness 1:__ This is a great comment and well taken. As you point out, it suffices to know $c_d, C_d, L$ just up to order without affecting rates. It is well-known \(f\) can be estimated in sup-norm with a KDE $\hat{f}$ at rate $||\hat{f} - f||_\infty \lesssim \left(\frac{N}{\log N}\right)^{-\frac{\alpha}{2\alpha+1}}$ with high probability using $N$ samples. This is actually much faster than we need since we only need the order of the unknown constants. For example, set $\hat{C}\_d := \max\_{|\mu| \leq 1} \hat{f}(\mu)$ and $\hat{c}\_d := \min\_{|\mu| \leq 1} \hat{f}(\mu)$, and note we have $\hat{C}\_d \asymp C\_d$ and $\hat{c}\_d \asymp c\_d$ with high probability, even if we only use a constant number of samples to fit $\hat{f}$. Similarly, $L$ can be estimated using a kernel density estimator $\hat{f}'$ of the derivative. Therefore, we can just siphon off a constant number of data points, estimate these constants up to order, and not affect the convergence rate.
>
> The question about unknown $\alpha$ is more delicate. Estimation of $\alpha$ itself is quite complicated, but perhaps the goal is instead  the construction of an adaptive estimator of $\mathcal{I}(f*\varphi_t)$ (i.e. does not require knowledge of $\alpha$) yet still achieves the minimax rate as if it were known.
>
> We believe it may be impossible to modify our methodology to be adaptive. The issue is we make use of a density point estimator in the low noise regime. Namely, we use an $\hat{f}$ with $E(|\hat{f}(\mu) - f(\mu)|^2) \lesssim n^{-\frac{2\alpha}{2\alpha+1}}$ for all $|\mu| \leq 1$. Let us fix a $\mu^* \in (-1, 1)$.
>
> It is a well known result due to Lepski (O. V. Lepskii. *On a problem of adaptive estimation in Gaussian white noise.* Theory of Probability \& Its Applications, 35(3):454-466, 1991) that $\hat{f}$ cannot achieve the minimax rate of estimating the density at $\mu^*$ over the class $\mathcal{F}\_{\alpha_1}$ and simultaneously over $\mathcal{F}\_{\alpha_2}$. In particular, it can be shown that for *any* estimator $\hat{f}$, if $\sup\_{f \in \mathcal{H}\_{\alpha_1}} E(|\hat{f}(\mu^*) - f(\mu^*)|^2) \lesssim n^{-\frac{2\alpha_1}{2\alpha_1+1}}$, then we actually have
> \begin{equation*}
>        \limsup_{n \to \infty} \sup_{f \in \mathcal{H}_{\alpha_2}} n^{\frac{2\alpha_2}{2\alpha_2 + 1}} E(|\hat{f}(\mu^*) - f(\mu^*)|^2) = \infty.
> \end{equation*}
> Therefore, our strategy seems doomed. It is very interesting to ask if some other approach can work. Thanks very much for pointing it out.
>
> __Weaknesses 2 and 3:__ Reviewer CeFj asked a question about empirics. Please see our response there. Thanks!

---

> > ### Comment · Reviewer_arrG · 2025-04-05
> >
> > Thanks to the authors for their detailed rebuttals. My main concerns have been addressed, and I have changed my recommendation to 3 (Weak Accept). The main limitation of the paper continues to be the lack of empirical results and some questions about how to actually implement this estimator in practice, so I think the paper would be much stronger if some experiments were added.
> >
> > I also think adding the discussion on the higher-dimensional case (from the rebuttal to Reviewer pEyQ) will strengthen the paper.

---

### Official Review · Reviewer_pEyQ · 2025-03-12

**Overall Recommendation:** 5

**Summary:**

The paper considers probability densities smoothed by Gaussian noise of variance $t$, and addresses the problem of estimating the Fisher information of the smoothed densities based on a collection of $n$ i.i.d. samples. The Fisher information can be expressed as an integral of the smoothed density and its derivative. The paper proposes a estimator, which is of the “plug-in” type, in the sense that first both the smoothed density and its derivative are estimated, and then plugged in into the Fisher information functional. The way in which the smoothed density and its derivative are proposed to be estimated depends on the variance of the smoothing Gaussian noise, and is different in three different regimes of $t$, and in general, based on properly truncated empirical PDF estimator, or on existing kernel-density estimators. The paper analyzes the expected error of these estimators. The paper also derive minimax lower bounds, which assert its rate-optimality, except in an intermediate regime for $t$ (not small enough, or not large enough), in which no lower bound is proved. The lower bounds are based on the two-point method (Le-Cam) with a proper choice of pair of densities – in the regime of low $t$, densities are sharp, and thus should be distinguishable at the edges of the support ($\pm 1$). The bound is thus based on a pair of distributions different at these edges. In the regime of high $t$, the densities are very smooth, and thus should be distinguished at their center (around $0$). The bound is thus based on a pair of distributions different at the center.

Then, using known identities (I-MMSE, de Bruijn), the estimator for the Fisher information is shown to be utilized to estimate the mutual information over the Gaussian channel (with the original density has the density of the input) and the output differential entropy. Both have $O(1/\sqrt{n})$ rates. Finally, some of the proof ideas are highlighted – perturbation analysis of the function determines the Fisher information (Propositions 5.1 and 5.2).

**Claims And Evidence:**

The paper claims that a simple plug-in achieves the minimax rate, with accurate dependency on the Gaussian smoothing variance $t$, then Holder-smoothness parameter of the density $\alpha$ and the number of samples $n$. The analysis of the estimator and the minimax lower bound validate this claim, up to a short interval in $t$, in which the question is open.

That being said, the proposed estimator is not a vanilla plug-in estimator. Indeed, the Fisher information is expressed as a function of the smoothed density and its derivative, and those quantities are estimated and plugged into the functional. However, the Fisher information is represented by different functionals in various regimes for $t$ (especially in the very high noise regime), and the way that the smoothed density and its derivative are estimated is different in each regime (possibly truncated, as smoothed empirical density or via a kernel density estimator). From my perspective the strong theoretical results of the paper are definitely of interest, but the estimator is not very simple.

**Essential References Not Discussed:**

I am not aware of an essential reference missing.

**Experimental Designs Or Analyses:**

Not applicable, the paper is purely theoretical.

**Methods And Evaluation Criteria:**

The evaluation method is standard and makes sense– minimax expected error rate for an estimator based on $n$ i.i.d. samples.

**Other Comments Or Suggestions:**

1) The name Fisher information is typically reserved for parametric families. Here it is somewhat hidden that the parametric family is the location. I think that in the ML community this quantity is referred to as Stein information.

2) Remark 1.2: How does it follow that $I_t \gtrsim 1/\sqrt{t}$ ?

3) In (6) and (7) there is a typo, I think that the integral should have $d\mu$.

4) If I understand correctly, the derivative of $\alpha$ Holder function will be $(\alpha-1)$ Holder. So in Section 2.2, for the estimate of the derivative, shouldn't the rate change to $n^{2(\alpha-1)/(2\alpha-1)}$ (replacing $\alpha$ with $\alpha-1$ in the preceding bound)?

5) In line 423, how $g(x,t)$ is defined? As a smoothed version of $g? A short explanation would clarify this.

**Other Strengths And Weaknesses:**

Strengths:

1) The paper is very well written with the main ideas and the merit of the results clearly explained. To the extent possible, the intuition of the proofs is explained.

2) The result is a sharp (almost full) characterization of the minimax estimation rate in this problem.

3) It is interesting that a plug-in estimator is optimal for $t>0$, as it is suboptimal in the unsmoothed case.

4) The estimator of the Fisher information results an estimator for other functionals – mutual information and differential entropy of the smoothed density.

Weaknesses:

1) High dimensions: The paper addresses one-dimensional densities. One of the main motivations for Gaussian smoothing is to circumvent the curse of dimensionality. It is not discussed anywhere in the paper (and is unclear) if the approach can be directly extended to high dimensions (with the anticipated technicalities), or if it breaks down at high dimensions.

2) The computational question is completely ignored, in the sense that it is not obvious how simple it is to compute the estimators – these are integrals of the estimated densities and the estimated derivatives of these densities over an infinite interval.

3) The paper is not a perfect fit to ICML, as it is purely theoretical, without any actual machine-learning applications, or even connection to machine learning techniques (the “information bottleneck” motivation is rather generic. This explains my recommendation is only “accept”.

**Questions For Authors:**

None

**Relation To Broader Scientific Literature:**

The problem of estimating statistical functionals is a classic problem, and the Fisher information is one of the central functionals in statistics. The paper addresses both classic works on the problem (without smoothing) and a recent line of work considering smoothed densities.

In a broader context, the motivation of the current work – Given that the Fisher information is discontinuous in $t$, and as the actual interest is in the Fisher information of the original density, why it is of interest to estimate it to begin with? The motivation questions also comes to mind given the fact that increasing $t$ from zero may actually make the problem more difficult.

Finally, given the interest in diffusion models, there are many papers addressing the problem of score estimation. As Fisher information is the variance of the score, it would be interesting to relate the paper more closely to this research area.

**Theoretical Claims:**

The theoretical claims are convincing, and I have verified the claims made in the paper, though the rigorous proofs are fully deferred to the appendix. From a quick overview of the appendix, and especially the techniques used, the proofs also appear to be convincing.

---

> ### Author Rebuttal · Authors · 2025-03-29
>
> Thanks for the constructive comments!
>
> __Weakness 1:__
> Suppose $f$ is an $\alpha$-Holder density on $[-1, 1]^d$. Our result can be directly extended, and the rate is entirely expected:
>
> $$\inf\_{\widehat{\mathcal{I}}\_t} \sup_{f \in \mathcal{F}\_\alpha} E\left(\left|\widehat{\mathcal{I}}\_t - \mathcal{I}\_t\right|\right) \asymp \frac{1}{\sqrt{n}t^2} \wedge \frac{1}{\sqrt{n} t^{(d+2)/4}} \wedge \frac{n^{-\alpha/(2\alpha+d)}}{\sqrt{t}}.$$
>
> This is natural from the well-known $d$-dimensional estimation rates of the plugged in density and gradient estimators.
>
> Notably, in the very high and high noise regimes, the convergence rate in terms of the sample size is the fast $\frac{1}{\sqrt{n}}$ rate; the curse of dimensionality is circumvented. A curse seems to appear in the low noise regime rate, but one needs to take care in the interpretation. Note the rate is the minimum of three terms, which means we *always* can achieve $\frac{1}{\sqrt{n}t^2}$. Therefore, one might say the curse never bites. But when $t$ is very small, one can beat $\frac{1}{\sqrt{n}t^2}$ and achieve the faster $\frac{n^{-\alpha/(2\alpha+d)}}{\sqrt{t}}$, which appears to suffer the curse.
>
> The answer to this apparent conceptual puzzle is that one cannot avoid paying $\frac{1}{t}$ raised to a power involving $d$ (in our case it is $t^{-\frac{d+2}{4}}$), which becomes large as $t$ gets small. This phenomenon is not unique to Fisher information estimation, and has been noted earlier by Goldfeld et al. (2020) that it occurs also in smoothed entropy estimation. Since some claim in these results that the curse of dimensionality has been circumvented, some might also claim that the curse is avoided in our problem too.
>
> Though we have not checked every single detail to confirm the conjectured rate, the generalization of the proof to the multivariate case seems very standard and only involving tedious notation. One might be worried about generalizing our use of integration by parts, but a coordinate-wise argument works as the domain is $[-1, 1]^d$. Since we use plug-in  the argument is quite similar to the score estimation theory of Dou et al. (2024).
>
> __Weakness 2:__ Reviewer CeFj commented on empirics, and in our response to them we have described how the estimators can be computed. Please have a look there. Thanks!
>
> __Relation to broader scientific literature:__ Reviewer SWHe pointed to some related work; please see our response!
>
> __Other Suggestion 1:__ Thanks! In the revision, we will make a note that this quantity is also known as the Stein information. After reading your comment, we were interested in completely replacing all instances of "Fisher information" with "Stein information", but after consulting some senior colleagues (from the nonparametric statistics and information theory communities and who also keep up with ICML/Neurips/etc), we decided to stick with the term "Fisher information" as the paper is particularly relevant to those communities. Thanks again for your comment, and we will be sure to note that it is also known as the Stein information.
>
> __Other Suggestion 2:__ Continuing the calculation from the remark, we have $\mathcal{I}\_t = \frac{1}{2}\int\_{-\infty}^{\infty} \frac{(\varphi_t(x+1) - \varphi_t(x-1))^2}{P(|N(x, t)| \leq t)} dx \gtrsim \int_{|x-1| \leq \sqrt{t}} \varphi_t(x-1)^2dx$ where the last inequality follows from $P(|N(x, t)| \leq 1) \asymp 1$ for $|x| \leq 1 + C\sqrt{t}$ as $t < 1$. We have also used that $\varphi_t(x+1) \leq c \varphi_t(x-1)$ for $|x-1| \leq \sqrt{t}$ where $c < 1$ is a small universal constant, since $t$ is small. Consider $\int_{|x-1| \leq \sqrt{t}} \varphi_t(x-1)^2 dx \asymp \frac{1}{\sqrt{t}} \int_{|x-1| \leq \sqrt{t}} \frac{1}{\sqrt{t}} e^{-\frac{(x-1)^2}{t}}dx \asymp \frac{1}{\sqrt{t}} P(|N(1, t) - 1| \leq \sqrt{t}) \asymp \frac{1}{\sqrt{t}}$. Hence, $\mathcal{I}_t \gtrsim \frac{1}{\sqrt{t}}$. In the revision, we will elaborate to make this clearer to the reader.
>
> __Other Suggestion 3:__ Thanks!
>
> __Other Suggestion 4:__ Though we agree the intuition is natural, it is an classic result from statistics that the minimax rate for estimating the $r$th derivative of an $\alpha$-Holder function in squared $L^2$ or squared pointwise error is $n^{-2(\alpha-r)/(2\alpha+1)}$ (we take $r = 1$ for our purposes). This result (along with results for other error metrics) is due to Charles Stone (namely, his papers *Optimal global rates of convergence for nonparametric regression*, The Annals of Statistics 10 (1982), no. 4, 1040-1053 and also *Optimal uniform rate of convergence for nonparametric estimators of a density function or its derivatives*, Recent Advances in Statistics, Elsevier, 1983, pp. 393-406). In our paper, we had only cited textbooks, but we will also cite these papers of Stone in the revision.
>
> __Other Suggestion 5:__ Thanks! Yes, $g(x, t) := (g*\varphi_t)(x)$. We had defined it on page 17, but neglected to point it out at line 423. We will make a note of it in the revision.

---

### Decision · Program_Chairs · 2025-05-01

**Decision:**

Accept (poster)

**Comment:**

This paper studies the nonparametric functional estimation problem for the Fisher information of $f * N(0, t)$ based on iid observations from $f$, where the density $f$ is assumed to be Holder-smooth of order at least 1 and bounded from both above and below. The minimax rate of estimation is worked out for this problem, which exhibits three regimes. Surprisingly, the simple plug-in approach could attain this minimax rate in all regimes.

All reviewers are unanimously positive about this paper and appreciate the technical significance, which I concur. Small concerns are also proposed, such as the requirement that the Holder smoothness parameter is at least 1, and the knowledge of the density upper and lower bounds in the estimator. Overall I am happy to recommend acceptance.

A minor comment: technically speaking Remark 1.2 is incorrect, as the Fisher info for the uniform distribution should be infinite, because the integral should be taken over the entire real line, not just $[-1, 1]$. Alternatively, one see that the Fisher information of a density is the Fisher information in the corresponding location model, while it is well-known that the uniform location model is not differentiable in quadratic mean because of the change on the support. The current Remark 1.2 also violates the data-processing inequality of the Fisher information.